# ON CONFORMAL MACHINE UNLEARNING

## ABSTRACT

The increasing demand for data privacy has made *machine unlearning (MU)* essential for removing the influence of specific training samples from machine learning models while preserving performance on retained data. However, most existing MU methods lack rigorous statistical guarantees or rely on heuristic metrics such as accuracy. To overcome these limitations, we introduce a new definition for MU based on *conformal prediction (CP)*, providing statistically sound, uncertainty-aware guarantees without the need for the concept of naive retraining. We formalize the proposed conformal criteria that quantify how often forgotten samples are excluded from CP sets, and propose empirical metrics to measure the effectiveness of unlearning. We further present a practical unlearning method designed to optimize these conformal metrics. Extensive experiments across diverse forgetting scenarios, datasets and models demonstrate the efficacy of our approach in removing targeted data.

## 1 INTRODUCTION

The accelerating deployment of data-driven technologies has underscored the need for models to evolve with changing knowledge and semantics.

In practice, entire categories of data can become obsolete, redefined, or deemed sensitive. For example, in safety-critical settings such as content moderation, classification taxonomies frequently evolve to reflect newly restricted or redefined classes of content. Similarly, as machine learning systems are increasingly used in e-commerce, recommendation, and inventory management, product categories often become obsolete or discontinued. In such settings, retraining from scratch to discard outdated information is often infeasible, highlighting the importance of *machine unlearning (MU)* as a mechanism for efficiently removing the influence of superseded classes and maintaining model reliability over time (Cao and Yang, 2015).

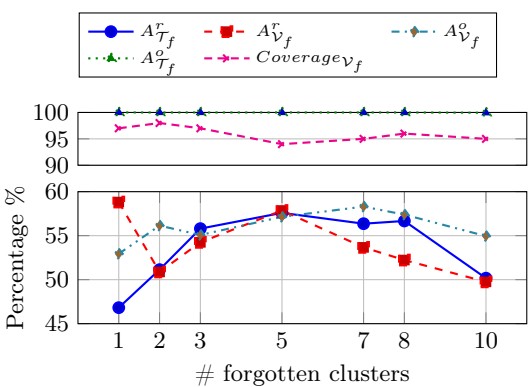

Figure 1: Plot of the train ($\mathcal{T}_f$) accuracy and validation ($\mathcal{V}_f$) accuracy and coverage over the forgotten data vs. the number of forgotten clusters in CIFAR100 for the retrained ($A^r$) and original ($A^o$) models.

Existing approaches fall into three main categories: (i) **Data-structure** methods that enable efficient partial retraining by leveraging specialized indexing or partitioning schemes (Ginart et al., 2019; Bourtoule et al., 2021); (ii) **Gradient-influence** and **variational** techniques that update model parameters to counteract the influence of forgotten samples (Warnecke et al., 2023; Graves et al., 2021; Nguyen et al., 2020); and (iii) **Knowledge-transfer** and **noise-based** objectives that aim to remove sensitive information while maintaining model utility (Chundawat et al., 2023a;b; Foster et al., 2024a).

Certified methods often draw on differential privacy (DP) (Dwork and Roth, 2014) or influence-function theory to provide formal guarantees on residual information leakage (Guo et al., 2020; Sekhari et al., 2021; Neel et al., 2021; Koh and Liang, 2017) (see Appendix C for more on related work). Compared to brute-force retraining, MU offers faster response times, reduced computational cost, and improved

| Metric | $A^r_{\mathcal{T}_r} \uparrow$ | $A^r_{\mathcal{T}_f} \downarrow$ | $A^r_{\mathcal{V}} \uparrow$ | $A^o_{\mathcal{V}} \uparrow$ |
|---|---|---|---|---|
| | 98.81 | 96.00 | 94.01 | 94.22 |

Table 1: Accuracy on forgotten and retained data for 100 samples random forgetting in AG-News. $A^r$ is accuracy from retrained model and $A^o$ from original model.

protection against membership-inference and model-inversion attacks (Shokri et al., 2017; Fredrikson et al., 2015). Despite these advances, three key limitations persist:

**L1 Certifiability relies on retrained baselines**: Most guarantees define unlearning as producing a model indistinguishable from one retrained from scratch without the forgotten data (Sekhari et al., 2021; Guo et al., 2020; Koloskova et al., 2025). This frames unlearning as merely *approximating* a retrained model, which both restricts objective design and makes assessment ambiguous, especially when the forgotten and validation data may not be identically distributed. For example, Fig. 1 demonstrates that accuracy on forgotten data can diverge from validation accuracy when the validation split is unrepresentative (gaps of $\approx 13\%$ for one forgotten cluster and $\approx 8\%$ for eight clusters). Consequently, approximate methods still *require* a retrained baseline to verify effectiveness—an approach that is computationally expensive and impractical at scale (Warnecke et al., 2023; Neel et al., 2021). There is a need for MU objectives that (i) flexibly reflect user requirements and (ii) can be certified or assessed via transparent, global measures rather than costly baselines.

**L2 Heuristic evaluation metrics**: Standard metrics (accuracy on forget/retain/test splits) neither quantify uncertainty nor provide clear guidance without a fully retrained reference (see **L1**). Because generalization effects make the impact of unlearning on forgotten or retained accuracy unpredictable, practitioners cannot tell whether true unlearning has occurred in some scenarios like cluster-wise forgetting. Moreover, these metrics lack uncertainty quantification. Fig. 1 demonstrates that even when a retrained model performs poorly on forgotten data (original train accuracy $\geq 99$ across clusters), conformal prediction at level $\alpha = 0.05$ still yields high coverage on those points. This *fake unlearning* phenomenon (Shi et al., 2025) persists even when accuracy drops are large (up to $\geq 40$ across different numbers of forgotten clusters), revealing a disconnect between accuracy and genuine forgetting and undermining the notion of the retrained model as a universal gold baseline for unlearning.

**L3 Random instance forgetting is unrealistic**: Unlearning performance evaluation by randomly deleting independent and identically distributed (i.i.d.) points does not reflect practical requests, which are typically *targeted* (e.g., removing all records for an individual, excising mislabeled samples, or excluding sensitive groups). Such deletions are structured and often correlated in feature or label space, whereas random removal mostly perturbs the training distribution and seldom shifts decision boundaries in a meaningful way (see Appendix J). Empirically, Table 1 shows negligible impact from random forgetting: accuracy on retained ($\mathcal{T}_r$) and forgotten ($\mathcal{T}_f$) sets is nearly identical, and test accuracy ($\mathcal{V}$) is unchanged. Thus, random instance forgetting is *not a meaningful or informative* evaluation scenario for machine unlearning.

These limitations raise a central question: *Can we define machine unlearning in a statistically rigorous, uncertainty-aware manner that eliminates the need for retrained baselines when evaluating effectiveness?*

We tackle this challenge by utilizing conformal prediction (CP) (Vovk et al., 2005; Angelopoulos et al., 2025; Angelopoulos and Bates, 2022; Lei and Wasserman, 2014) to develop MU with guarantees that do not depend on retraining:

- **Unified unlearning objectives.** CP lets practitioners set explicit uncertainty targets via the coverage level $\alpha$. By jointly controlling miscoverage on forgotten data, we obtain a clear, model-agnostic objective that unifies evaluation under a common uncertainty-quantification framework.

- **Statistical guarantees.** CP yields finite-sample coverage guarantees under mild exchangeability. Extending these to constrain miscoverage on forgotten data while preserving coverage on retained data provides a rigorous basis for MU.

Our main contributions are summarized as follows:

- **New MU definition.** A conformal-probability objective quantifies forgetting without comparison to retrain-from-scratch models.
- **Theory and metrics.** We derive $(\alpha, \beta)$-conformal guarantees and introduce practical metrics—*ECF* at $c$ and *EmCF* at $d$—for assessing unlearning.
- **Empirical method.** We propose a scalable algorithm aligned with these guarantees, achieving strong unlearning while maintaining performance on retained data.

Our perspective moves beyond DP-based definitions, offering a conformal, probabilistic foundation that is both principled and practical. The first work to motivate MU via CP is (Shi et al., 2025), which studies *fake unlearning* and proposes CR/MIACR metrics and a corresponding method; however, CR does not fully resolve fake unlearning (see Appendix C).

## 2 PRELIMINARIES AND NOTATIONS

Let $X \in \mathcal{X}$ and $Y \in \mathcal{Y}$ denote features and labels, and write $\mathcal{D} \subset \mathcal{X} \times \mathcal{Y}$ with $\mathcal{D} \sim p_{X,Y}$. A training set $\mathcal{D}_{\text{train}} \sim p_{X,Y}$ produces a model $f_{\theta_o}$ with parameters $\theta_o$. A MU algorithm $\mathfrak{U}$ modifies $\theta_o$ to $\theta_u$, yielding $f_{\theta_u}$ that *retains* performance on $\mathcal{D}_r$ while *forgetting* $\mathcal{D}_f$ (our formal objective is in Section 3). We focus on *targeted forgetting*. Let a target variable $W \in \mathcal{W}$ encode the characteristics to be forgotten; the joint admits the mixture

$$p_{X,Y}(x,y) = \int_{\mathcal{W}} p_{X,Y|W}(x,y \mid w) \, p_W(w) \, \mathrm{d}w, \tag{1}$$

with

$$\mathcal{D}_f \sim p_{X,Y|W}(\cdot, \cdot \mid W \in \mathcal{W}_{\text{forget}}), \qquad \mathcal{D}_r \sim p_{X,Y|W}(\cdot, \cdot \mid W \in \mathcal{W} \backslash \mathcal{W}_{\text{forget}}), \tag{2}$$

and $\mathcal{D}_{\text{unlearn}} = \mathcal{D}_f \cup \mathcal{D}_r$. Examples include class-level forgetting ($W = Y$) and feature- or subspace-based criteria ($W = X$ or $W = \Pi X$, where $\Pi$ is a projection operator). We assume that, conditioned on $\mathcal{D}_{\text{train}}$, the points in $(\mathcal{D}_{\text{unlearn}}, \mathcal{D}_{\text{calib}})$ are exchangeable (Angelopoulos et al., 2025). The common "random forgetting" setup (Foster et al., 2024b;a; Shi et al., 2025; Peng et al., 2025; Graves et al., 2021) is a special case where $\mathcal{D}_f$ is a random sub-sample of $\mathcal{D}_{\text{train}}$ (see Appendix J for why this is typically uninformative for practice).

Our framework leverages CP to provide both evaluation and training signals: intuitively, we seek to ensure that the conformal set $C(X; \theta_u)$ *rarely* covers points from $\mathcal{D}_f$, while maintaining user-specified coverage on $\mathcal{D}_r$. The formal definitions are given in Section 3 and the overall workflow is illustrated in Fig. 2. For comparison, we also consider a *retrained model* $f_{\theta_{\text{RT}}}$, which is trained from scratch on $\mathcal{D}_r$ with parameters $\theta_{\text{RT}}$. A summary of the conformal prediction background relevant to this work is provided in Appendix B. Table 6 in Appendix A.1 provides a summary of commonly used notations.

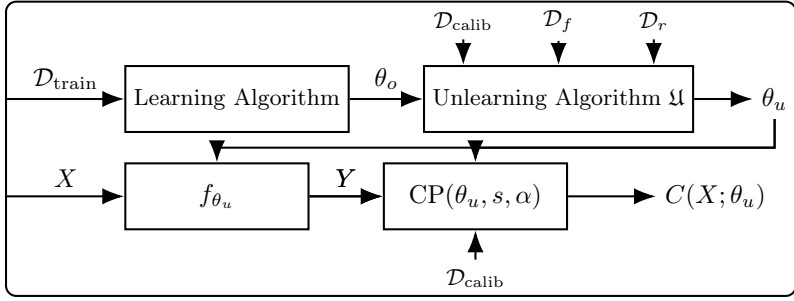

Figure 2: Conformal MU workflow.

## 3 Conformal Machine Unlearning

In this section, we introduce a new notion of conformal unlearning grounded in CP theory. We formalize the conditions under which conformal unlearning provides statistical guarantees. We propose empirical metrics to assess the performance of MU algorithms, and present a practical conformal unlearning framework informed by our theoretical analysis. Proofs of theoretical results are deferred to Appendix D.

### 3.1 Definitions and Theory

Consider a machine learning model $f_{\theta_o}$ and a CP procedure $\mathrm{CP}(\theta_o, s, \alpha)$. A MU algorithm $\mathfrak{U}$ transforms the model parameters $\theta_o$ into $\theta_u = \mathfrak{U}(\theta_o, \mathcal{D}_f)$, where $\theta_u$ are the model parameters after unlearning the forget set $\mathcal{D}_f$.

**Definition 1** (($\alpha, \beta$)-conformal unlearning). *An unlearning algorithm $\mathfrak{U}$ is said to be ($\alpha, \beta$)-conformal unlearning for $0 \leq \alpha \leq \beta \leq 1$, if*

$$\mathbb{P}(Y \in C(X; \theta_u) \,|\, (X, Y) \in \mathcal{D}_r) \geq 1 - \alpha, \tag{3}$$

$$\mathbb{P}(Y \notin C(X; \theta_u) \,|\, (X, Y) \in \mathcal{D}_f) \geq \beta. \tag{4}$$

In the MU framework given by Definition 1, $\alpha$ is the user-defined miscoverage rate inherent to CP. Its value is application-dependent. Once the user sets this tolerance level, the goal of the unlearning method is to maximize $\beta$. On the other hand, $\beta$ quantifies the strength of unlearning: it is the probability that forgotten data are excluded from the model's prediction sets. In sensitive applications such as unlearning confidential or regulated information, a large $\beta$ is essential to minimize the risk of information leakage. In less critical contexts, a lower $\beta$ may be acceptable; for example, if the goal is to forget a broad category, occasional inclusion in a prediction set may not be consequential.

The parameter $\beta$ is the primary parameter of focus for evaluating unlearning. Our framework leverages the standard $\alpha$ from CP to ensure that the model includes the true labels of retained data in its prediction sets. This framework offers a new paradigm for machine unlearning. Rather than being limited to approximating a retrained model, we have proposed a more flexible, statistically-grounded objective. This empowers users to define their unlearning goals in a way that is tailored to their specific needs and risk tolerance.

**Corollary 1.** *In targeted forgetting, suppose $\mathbb{P}((X, Y) \in \mathcal{D}_r) > 0$. Then, (4) implies (3).*

On the other hand, suppose (3) holds. In the targeted forgetting scenario, if the conformity scores of the data samples in $\mathcal{D}_{\mathrm{calib}}$ are continuous and do not have any ties (Lei et al., 2018) (which is always achievable by adding a small random noise), then letting $n = |\mathcal{D}_{\mathrm{calib}}|$, $\delta = \mathbb{P}((X, Y) \in \mathcal{D}_f) > 0$ and $\widetilde{\beta} = \mathbb{P}(Y \notin C(X; \theta_u) \,|\, (X, Y) \in \mathcal{D}_f)$, we have

$$1 - \alpha + \frac{1}{n+1} \geq \mathbb{P}(Y \in C(X; \theta_u)) \geq (1-\alpha)(1-\delta) + (1-\widetilde{\beta})\delta \implies \widetilde{\beta} \geq \alpha - \frac{1}{(n+1)\delta},$$

which does not guarantee (4) for $\beta \geq \alpha$. This indicates that the MU algorithm has to be non-trivial to satisfy Definition 1.

**Corollary 2.** *In targeted forgetting, suppose (20) and (4) hold. Then,*

$$\mathbb{P}((X, Y) \in \mathcal{D}_f) \cdot \beta \leq \alpha \leq \beta. \tag{5}$$

Any MU algorithm that achieves (3) and (4) for relatively small $\alpha$ and large $\beta$ has a statistically strong unlearning property. For a calibration set $\mathcal{D}_{\mathrm{calib}}$ exchangeable with $\mathcal{D}_{\mathrm{unlearn}}$, the worst ($\alpha, \beta$)-conformal unlearning method operating on $\mathcal{D}_{\mathrm{calib}}$ is given by $\alpha = \frac{1}{n+1}$ since any smaller value of $\alpha$ leads to $\hat{q}_\alpha = \infty$ and $C(X; \theta_u) = \mathcal{Y}$ for all $X$. Hence, the unlearning collapses, and all samples become covered with the trivial set. In that case, the forget set miscoverage probability in (4) is 0, and is excluded from Definition 1. In another trivial case when $\alpha = 1$, then Corollary 2 gives $\beta = 1$, which is expected.

In the case of *zero-shot* MU (as defined in Foster et al. (2024a)) where sampling from the same distribution as the forget set $\mathcal{D}_f$ is not available, we are restricted to a calibration set whose samples are from the same underlying distribution as the retain set $\mathcal{D}_r$. We have the following result.

**Proposition 1.** *Suppose $\mathcal{D}_{\text{calib}}$ and $\mathcal{D}_r$ are exchangeable, and are independent of $\mathcal{D}_f$. Let $(X_f, Y_f) \in \mathcal{D}_f$ and $(X_r, Y_r) \in \mathcal{D}_r$. A $(\alpha, \beta)$-conformal unlearning algorithm $\mathfrak{U}$ based on $\mathcal{D}_{\text{calib}}$ and $\mathcal{D}_r$ yields*

$$\mathbb{P}(s(X_f, Y_f; \theta_u) \geq s(X_r, Y_r; \theta_u)) \geq \beta(1 - \alpha), \tag{6}$$

*i.e., the unlearned model $f_{\theta_u}$ assigns a worse conformity score to the samples from $\mathcal{D}_f$ than to samples from $\mathcal{D}_r$ with high probability for small $\alpha$ and large $\beta$.*

A conformal prediction set identifies the most likely labels for a test sample. In practice, large prediction sets become uninformative, which is undesirable for prediction on $\mathcal{D}_r$ but desirable for $\mathcal{D}_f$. Therefore, we also consider the following relaxed version of Definition 1 where the coverage or miscoverage are restricted to efficient (small) prediction sets.

**Definition 2** ($(c, d)$-efficient $(\alpha, \beta)$-conformal unlearning). *An unlearning algorithm $\mathfrak{U}$ is said to be $(c, d)$-efficient $(\alpha, \beta)$-conformal unlearning for $0 \leq \alpha \leq \beta \leq 1$ and integers $c, d \in \{0, \ldots, |\mathcal{Y}|\}$, if*

$$\mathbb{P}(Y \in C(X; \theta_u) \mid (X, Y) \in \mathcal{D}_r, |C(X; \theta_u)| \leq c) \geq 1 - \alpha, \tag{7}$$

$$\mathbb{P}(Y \notin C(X; \theta_u) \mid (X, Y) \in \mathcal{D}_f, |C(X; \theta_u)| \leq d) \geq \beta. \tag{8}$$

The thresholds $c$ (retained) and $d$ (forgotten) specify the largest prediction-set sizes deemed *informative* by the user. Sets exceeding these limits are treated as "inefficient" and *excluded* from coverage calculations: they are too broad to be actionable, and coverage achieved only with such large sets reflects low model confidence (Shafer and Vovk, 2008). The choice of $c, d$ is application-specific and encodes the user's uncertainty tolerance.

To illustrate, consider a 20-way document classification. If the model returns a set of size 10 that includes a confidential (to-be-forgotten) label, the set is too broad to constitute a meaningful leak; a size-5 set containing that label *is* informative and would count against forgetting. For retained labels, a size-7 set still usefully narrows the group of potential true labels. A practitioner might thus choose $d=5$ (forgotten) and $c=7$ (retained). In practice, evaluating several $(c, d)$ pairs clarifies trade-offs. When $c=d=|\mathcal{Y}|$, Definition 2 reduces to standard $(\alpha, \beta)$-conformal unlearning.

**Corollary 3.** *Under the same assumptions as Corollary 1, suppose a MU algorithm $\mathfrak{U}$ is $(\alpha, \beta)$-conformal unlearning. For $c, d \in \{0, \ldots, |\mathcal{Y}|\}$, assume $\mathbb{P}(|C(X; \theta_u)| > c \mid (X, Y) \in \mathcal{D}_r) \leq \zeta_c$, and $\mathbb{P}(|C(X; \theta_u)| > d \mid (X, Y) \in \mathcal{D}_f) \leq \eta_d$. Then, we have*

$$\mathbb{P}(Y \in C(X; \theta_u) \mid (X, Y) \in \mathcal{D}_r, |C(X; \theta_u)| \leq c) \geq 1 - \alpha - \zeta_c, \tag{9}$$

$$\mathbb{P}(Y \in C(X; \theta_u) \mid (X, Y) \in \mathcal{D}_f, |C(X; \theta_u)| \leq d) \geq \beta - \eta_d. \tag{10}$$

Corollary 3 indicates that by having a sufficiently good model so that $\zeta_c$ and $\eta_d$ are small, we can achieve a good bound on the coverage of the retained points and the miscoverage of the forgotten points.

In the targeted forgetting scenario considered in this work, we assume that $\mathcal{D}_{\text{calib}}$ and $\mathcal{D}_{\text{unlearn}}$ are exchangeable (conditioned on $\mathcal{D}_{\text{train}}$). This exchangeability is crucial for the coverage guarantees in (20), (3), and (4). In practice, exchangeability is easily achieved for class-wise and group-wise unlearning by holding out validation points for each class or group during training. However, in instance-wise unlearning, the forgotten data points are typically used for both training and unlearning, which breaks exchangeability and invalidates the coverage guarantee in (3) unless the calibration set is chosen by resampling from $\mathcal{D}_{\text{train}}$. However, that cannot be guaranteed if downstream users receive the unlearned model without knowledge of the unlearning procedure/request. Our aim is to develop empirical machine unlearning metrics that closely mirror both (3) and (4), and remain robust even when exchangeability is only approximately satisfied.

If exchangeability does not hold, more general forms of conformal prediction can be used, such as the non-exchangeable CP framework with coverage gap correction (Foygel Barber et al., 2023). In this setting, the coverage gap must be accounted for in all relevant guarantees, including (20), (3), and (4). We discuss the non-exchangeable scenario in detail in Appendix F.

## 4 Conformal Unlearning Metrics and Framework

In this section we introduce the *Efficiently Covered Frequency (ECF at c)* and its complement, the *Efficiently Miscovered Frequency (EmCF at d)* metrics as surrogates for informative coverage/miscoverage. In addition, we present the Conformal Quantile-based Machine Unlearning (CQMU) framework, a practical unlearning method aimed at maximizing these metrics.

### 4.1 Empirical Conformal Unlearning Metrics

In this subsection, we introduce new empirical metrics that help practitioners evaluate their MU algorithms' performance in an uncertainty-quantified manner *without the need for the naively-retrained baseline.*

Following Definition 1, a good conformal unlearning algorithm should maximize the empirical version of both left-hand sides (L.H.S.) of (3) and (4). From Definition 2, as far as CP is concerned, large prediction sets become quickly uninformative. Therefore, large prediction sets also indicate low confidence of the model in classifying those points. Hence, we introduce a relaxed version of the coverage metric in this work. As a result, we want to maximize the L.H.S. of (7) and (8).

We approximate probabilities by frequencies (Shi et al., 2025). For a retain set $\mathcal{D}_r$ and positive integer $c$, we define the *efficiently covered frequency (ECF) at c* of a prediction set $C(\cdot)$ as

$$\mathfrak{C}_{\mathcal{D}_r}(c) = \frac{1}{|\mathcal{D}_r(c)|} \sum_{(x,y)\in\mathcal{D}_r(c)} \mathbf{1}\{y \in C(x)\}, \tag{11}$$

where $\mathcal{D}_r(c) = \{(x,y) \in \mathcal{D}_r : |C(x)| \leq c\}$ and $\mathbf{1}\{\cdot\}$ is the indicator function.

On the other hand, for a forgotten dataset $\mathcal{D}_f$ and positive integer $d$, we define the *efficiently miscovered frequency (EmCF) at d* of a prediction set $C(\cdot)$ as

$$\mathfrak{M}_{\mathcal{D}_f}(d) = \frac{1}{|\mathcal{D}_f(d)|} \sum_{(x,y)\in\mathcal{D}_f(d)} \mathbf{1}\{y \notin C(x)\}, \tag{12}$$

where $\mathcal{D}_f(d) = \{(x,y) \in \mathcal{D}_f : |C(x)| \leq d\}$.

Note that $C(\cdot)$ is the same in the whole framework. In our experiments, we test different values of $c$ and $d$ (see Appendix H.2). The final objective is to maximize both (11) and (12) for $\mathcal{D}_f$. Note that although $\mathfrak{C}$ and $\mathfrak{M}$ are mathematically simple extensions of the coverage, they correspond to significant statistical interpretation as illustrated after Definition 2.

### 4.2 Conformal Unlearning-Based Framework

From (20), a point is covered iff its non-conformity score is at most the calibration quantile: $s(X,Y;\theta) \leq \hat{q}_\alpha$. Thus, to *forget* we should push forgotten scores *above* $\hat{q}_\alpha$, while to *retain* we should keep retained scores *below* $\hat{q}_\alpha$. Our empirical goal—maximizing the coverage/miscoverage metrics in (11) and (12)—therefore reduces to shaping the *margins* of scores around the quantile. Concretely, we minimize a surrogate of the gap between forgotten scores and $\hat{q}_\alpha$ (pushing up) and the gap between retained scores and $\hat{q}_\alpha$ (pulling down), yielding the following unlearning loss (see Appendix E for a derivation):

$$\widetilde{L}(\theta_u; \mathcal{D}_f, \mathcal{D}_r) = -\widetilde{\varepsilon}_f(\theta_u) - \widetilde{\varepsilon}_r(\theta_u), \tag{13}$$

where

$$\widetilde{\varepsilon}_r(\theta_u) = \frac{1}{|\mathcal{D}_r|} \sum_{(X,Y)\in\mathcal{D}_r} \ell((1+\delta)p_{\theta_u}(Y_q \mid X_q) - p_{\theta_u}(Y \mid X)), \tag{14}$$

$$\widetilde{\varepsilon}_f(\theta_u) = \frac{1}{|\mathcal{D}_f|} \sum_{(X,Y)\in\mathcal{D}_f} \ell((1+\delta)p_{\theta_u}(Y \mid X) - p_{\theta_u}(Y_q \mid X_q)), \tag{15}$$

and $(X_q, Y_q) \in \mathcal{D}_{\text{calib}}$ represents the calibration point whose conformity score is closest to $\hat{q}_\alpha$ and its label, and $p_{\theta_u}(\cdot)$ is the softmax probability. The function $\ell(\cdot)$ is the sigmoid surrogate to the indicator function with steepness factor $\gamma$ (not shown explicitly above), and $\delta$ is a margin buffer that we choose to be $\delta = 0.0001$ in all our experiments.

The objective is to minimize (13). Since we typically initialize from the original trained model, we incorporate this by adding a regularization term that penalizes deviation from the original weights. This helps prevent significant drops in performance on the retained classes and mitigates overfitting to the specific forget and retain sets. The final optimization objective is therefore

$$\widehat{\theta}_{o \to u} = \arg\min_{\theta \in \Theta} \left\{ \widetilde{L} + \lambda \left\| \theta - \widehat{\theta}_o \right\|_2^2 \right\}. \tag{16}$$

Our framework introduces three auxiliary hyperparameters: the regularization weight $\lambda$, the margin buffer $\delta$, and the surrogate steepness $\gamma$. We use an $\ell_2$ penalty for differentiability; other norms are possible. The buffer $\delta$ stabilizes scores by nudging them beyond the quantile to avoid hard thresholding (as in Shi et al. (2025)); we *do not* tune $\delta$ and fix it to $10^{-4}$ in all experiments. Thus, beyond learning rate and epochs, only *two* hyperparameters $(\lambda, \gamma)$ require tuning. We refer to our method as *Conformal Quantile-based MU (CQMU)*. It does not require access to the original training set $\mathcal{D}_{\text{train}}$; it only needs $\mathcal{D}_{\text{calib}}$, $\mathcal{D}_f$, and a subset of retained points $\mathcal{D}_r$ with $\mathcal{D}_f \cup \mathcal{D}_r$ exchangeable with $\mathcal{D}_{\text{calib}}$. This makes CQMU practical when $\mathcal{D}_{\text{train}}$ is unavailable (e.g., third-party deployments). A full summary appears in Algorithm 1.

## 5 Numerical Experiments

**Datasets and models.** We evaluate on CIFAR-100 (Krizhevsky, 2009), a 100-class Tiny-ImageNet subset we call Imagenet100 (Shekhar, 2021), and 20 NewsGroups (20 classes) (Lang, 1995) (see Appendix G.1). Vision models use ResNet18; text models use BERTa-Distill. Unless noted, results are averaged over 6 random seeds.

**Training and unlearning.** CIFAR-100 uses SGD for 50 epochs (lr $0.1 \to 10^{-4}$ linear decay, momentum 0.9, wd $5 \times 10^{-4}$); the Tiny-ImageNet subset uses the same setup for 80 epochs. Text models train for 15 epochs (initial lr 0.01). Batch size is 256 with 2 workers; standard normalization/transforms are applied. Our unlearning optimizer matches the base (same momentum and wd) with a tuned learning rate and no scheduler. Experiments run on four NVIDIA RTX A5000 GPUs with `nn.DataParallel`.

**Data partitions.** We consider six subsets for evaluation: training forget/retain $(\mathcal{T}_f, \mathcal{T}_r)$, unlearning forget/retain $(\mathcal{D}_f, \mathcal{D}_r)$, and unseen forget/retain $(\mathcal{V}_f, \mathcal{V}_r)$. For Imagenet100, training uses 117k images; 13k are held out for unlearning/testing (6.5k calibration $\mathcal{D}_{\text{calib}}$ and 6.5k split into $\mathcal{V}_f, \mathcal{V}_r$). The 5k test split is used to form a 4k testing calibration set and label-based $\mathcal{D}_f, \mathcal{D}_r$ for unlearning. Baselines that require validation reuse $\mathcal{D}_{\text{calib}}$ (see Appendix H.1).

**Baselines.** We compare to $\nabla\tau$ (Trippa et al., 2024), SCRUB (Kurmanji et al., 2023), SSD (Foster et al., 2024b), AMN (Graves et al., 2021), BADT (Chundawat et al., 2023a), and UNSIR (Tarun et al., 2024), plus a retrained model (RT) on $\mathcal{T}_r$. Implementations use authors' code and CIFAR-100 hyperparameters from (Foster et al., 2024b; Chundawat et al., 2023a); we perform grid-search fine-tuning per method. Our paradigm applies unlearning on $\mathcal{D}_f/\mathcal{D}_r$ (not $\mathcal{T}_f/\mathcal{T}_r$).

**Metrics.** We report: $A_\mathcal{D}$ (accuracy percentage on $\mathcal{D} \in \{\mathcal{D}_r, \mathcal{D}_f, \mathcal{T}_r, \mathcal{T}_f, \mathcal{V}_r, \mathcal{V}_f\}$), $\mathfrak{C}_\mathcal{D}(c)$ on retained subsets $(\mathcal{D}_r, \mathcal{T}_r, \mathcal{V}_r)$, $\mathfrak{M}_\mathcal{D}(d)$ on forgotten subsets $(\mathcal{D}_f, \mathcal{T}_f, \mathcal{V}_f)$ with $c=d$, the harmonic mean $H$ over these six conformal metrics ($H = n / \sum_i x_i^{-1}$; $H=0$ if any $x_i=0$), MIA Diff (attacker accuracy percentage minus majority-class ratio), and Tsec (unlearning time in seconds). Imagenet results appear in the main text; extended vision and all text results are in Appendix H.2.

## 6 RESULTS AND DISCUSSION

For targeted class-wise unlearning, we use $\mathcal{D}_f$ and $\mathcal{D}_r$ as the unlearning sets. CQMU requires computing the CP quantile at each epoch, for which we use $\mathcal{D}_{\text{calib}}$. For baseline methods, validation forget and retain sets are required; we construct these as appropriate label-based subsets of $\mathcal{D}_{\text{calib}}$.

In Table 2, the values corresponding to $\mathfrak{C}$, $\mathfrak{M}$ represent coverage/miscoverage rates. If $\mathfrak{C} \geq 1 - \alpha$ or $\mathfrak{M} \geq \alpha$ (we desire $\beta >> \alpha$) we use a $\bullet$ to mark the result. RT results are included for reference. Note that for class-wise unlearning, RT is expected to be the optimal baseline since it does not output the forgotten labels. Table 12 in Appendix H.2 demonstrates that RT suffers from the fake unlearning problem for cluster-wise unlearning.

Table 2: Imagenet100 targeted class-wise forgetting coverage/miscoverage results with $c, d = 40$, $\alpha = 0.05$, and 10 forgotten class.

| Metric | $\mathfrak{C}_{\mathcal{D}_r}(c)$ ↑ | $\mathfrak{M}_{\mathcal{D}_f}(d)$ ↑ | $\mathfrak{C}_{\mathcal{T}_r}(c)$ ↑ | $\mathfrak{M}_{\mathcal{T}_f}(d)$ ↑ | $\mathfrak{C}_{\mathcal{V}_r}(c)$ ↑ | $\mathfrak{M}_{\mathcal{V}_f}(d)$ ↑ | $H$ ↑ |
|---|---|---|---|---|---|---|---|
| RT | $\bullet$ 0.96 ± 0.01 | $\bullet$ 1.00 ± 0.00 | $\bullet$ 1.00 ± 0.00 | $\bullet$ 1.00 ± 0.00 | $\bullet$ 0.98 ± 0.01 | $\bullet$ 1.00 ± 0.00 | 0.99 ± 0.00 |
| $\nabla\tau$ | $\bullet$ 1.00 ± 0.00 | $\bullet$ 0.16 ± 0.08 | $\bullet$ 0.99 ± 0.00 | 0.03 ± 0.01 | $\bullet$ 0.96 ± 0.00 | $\bullet$ 0.12 ± 0.03 | 0.10 ± 0.04 |
| SCRUB | $\bullet$ 1.00 ± 0.00 | 0.04 ± 0.01 | $\bullet$ 1.00 ± 0.00 | 0.00 ± 0.00 | $\bullet$ 0.96 ± 0.00 | 0.03 ± 0.00 | 0.00 ± 0.00 |
| SSD | 0.00 ± 0.00 | 0.00 ± 0.00 | 0.00 ± 0.00 | 0.00 ± 0.00 | 0.00 ± 0.00 | 0.00 ± 0.00 | 0.00 ± 0.00 |
| AMN | $\bullet$ 1.00 ± 0.00 | $\bullet$ 0.98 ± 0.01 | $\bullet$ 1.00 ± 0.00 | 0.00 ± 0.00 | $\bullet$ 0.97 ± 0.00 | $\bullet$ 0.10 ± 0.01 | 0.00 ± 0.00 |
| BADT | 0.92 ± 0.01 | $\bullet$ 0.19 ± 0.09 | $\bullet$ 1.00 ± 0.00 | 0.01 ± 0.00 | $\bullet$ 0.97 ± 0.00 | $\bullet$ 0.13 ± 0.02 | 0.06 ± 0.02 |
| UNSIR | $\bullet$ 1.00 ± 0.00 | 0.00 ± 0.00 | $\bullet$ 1.00 ± 0.00 | 0.00 ± 0.00 | $\bullet$ 0.99 ± 0.00 | 0.00 ± 0.00 | 0.00 ± 0.00 |
| CQMU (ours) | $\bullet$ 0.97 ± 0.00 | $\bullet$ 0.92 ± 0.03 | $\bullet$ 1.00 ± 0.00 | $\bullet$ 0.40 ± 0.02 | $\bullet$ 0.97 ± 0.00 | $\bullet$ 0.69 ± 0.02 | 0.74 ± 0.01 |

Table 2 evaluates whether the goal $\mathfrak{C} \geq 1 - \alpha$ is achieved on retained data and $\mathfrak{M} \gg \alpha$ on forgotten data. $\nabla\tau$ achieves high retained coverage ($\geq 0.95$) but fails to miscover $\mathcal{T}_f$ ($c=40$, $\beta \leq \alpha = 0.05$). SCRUB exhibits a similar pattern, suggesting that KL-based external-data objectives may be insufficient for effective forgetting. SSD collapses to large prediction sets on external data (all $\mathfrak{C}/\mathfrak{M}=0$ at $c=40$) despite hyperparameter tuning, likely due to similar parameter importance scores on unseen data. AMN overfits to $\mathcal{D}_f$ (miscoverage $\approx 0.98$) but fails to miscover $\mathcal{T}_f$ ($\approx 0.00$), while still covering retained data well. BADT and UNSIR degrade retained performance, resulting in inflated set sizes; UNSIR fails to miscover forgotten points, and BADT under-covers $\mathcal{D}_r$ ($0.92 < 0.95$) with only moderate forgotten miscoverage. In contrast, CQMU consistently satisfies $\mathfrak{C} \geq 0.95$ and $\mathfrak{M} \geq 0.40 \gg 0.05$ across all subsets, achieving a substantial $H$ margin over the next best method, other than RT. These results support our thesis that global, coverage-based criteria can replace costly retrained baselines while enabling uncertainty-aware evaluation.

Table 3 illustrates the phenomenon of *fake unlearning* (cf. Section 1): several methods show substantial accuracy drops on the training-forgotten split $\mathcal{T}_f$—for example, $\nabla\tau$ ($\geq 50\%$), AMN ($\geq 30\%$), and BADT ($\geq 50\%$)—yet their miscoverage $\mathfrak{M}$ on the same points remains below $\alpha$ (see also Table 2). This means that even when accuracy is low, the conformal set $C(X; \theta_u)$ still frequently contains the true label, so these samples are *covered* rather than *miscovered*. This discrepancy arises because accuracy penalizes any top-1 error, while conformal coverage only requires the true label to appear anywhere in the prediction set; thus, methods that simply widen prediction sets (or fail to shrink them sufficiently, e.g., at $c=40$) can exhibit large accuracy drops without achieving true forgetting. These findings reinforce our argument from Section 1: **accuracy alone cannot reliably indicate genuine unlearning** and may instead reflect *fake* forgetting. In contrast, CQMU aligns accuracy drops on forgotten data with a corresponding increase in $\mathfrak{M}$ (i.e., true labels are excluded from $C(X; \theta_u)$ at the target set size), while maintaining retained coverage ($\mathfrak{C} \geq 1-\alpha$), thus satisfying Definition 2. This coupling between accuracy and $\mathfrak{M}$ demonstrates that CQMU does not simply degrade logits indiscriminately, but instead shapes prediction sets so that forgotten concepts are genuinely unsupported, while retained concepts remain reliably covered.

While CQMU performs unlearning using external $\mathcal{D}_f$, it also effectively reduces coverage on $\mathcal{T}_f$—as expected in targeted forgetting. Because conditioning on $W$ induces a shared structure, $\mathcal{D}_f$ and $\mathcal{T}_f$ occupy similar regions in feature space; thus, shifting decision boundaries to miscover $\mathcal{D}_f$ naturally impacts $\mathcal{T}_f$. The observed increase in $\mathfrak{M}$ and the corresponding drop in accuracy on both forgotten splits demonstrate the intended effect of *targeted* unlearning.

Table 3: Imagenet100 targeted class-wise forgetting accuracy results with $c, d = 40$, $\alpha = 0.05$, and 10 forgotten class.

| Metric | $A_{\mathcal{T}_r} \uparrow$ | $A_{\mathcal{T}_f} \downarrow$ | $A_{\mathcal{D}_r} \uparrow$ | $A_{\mathcal{D}_f} \downarrow$ | $A_{\mathcal{V}_r} \uparrow$ | $A_{\mathcal{V}_f} \downarrow$ |
|---|---|---|---|---|---|---|
| Original | $99.74 \pm 0.00$ | $99.52 \pm 0.00$ | $66.45 \pm 0.00$ | $72.73 \pm 0.00$ | $78.25 \pm 0.00$ | $69.90 \pm 0.00$ |
| RT | $99.60 \pm 0.03$ | $0.00 \pm 0.00$ | $66.96 \pm 0.52$ | $0.00 \pm 0.00$ | $71.38 \pm 0.05$ | $0.00 \pm 0.00$ |
| $\nabla\tau$ | $80.25 \pm 5.25$ | $48.35 \pm 7.20$ | $77.39 \pm 4.60$ | $27.84 \pm 4.39$ | $55.55 \pm 3.05$ | $31.37 \pm 3.87$ |
| SCRUB | $99.52 \pm 0.00$ | $99.76 \pm 0.03$ | $89.67 \pm 0.15$ | $61.93 \pm 1.43$ | $70.19 \pm 0.11$ | $73.25 \pm 0.42$ |
| SSD | $1.12 \pm 0.01$ | $0.00 \pm 0.00$ | $1.30 \pm 0.04$ | $0.00 \pm 0.00$ | $1.01 \pm 0.14$ | $0.00 \pm 0.00$ |
| AMN | $99.43 \pm 0.02$ | $71.61 \pm 4.92$ | $100.00 \pm 0.00$ | $0.00 \pm 0.00$ | $66.91 \pm 0.30$ | $26.19 \pm 3.49$ |
| BADT | $98.42 \pm 0.43$ | $51.26 \pm 5.75$ | $64.22 \pm 0.39$ | $9.28 \pm 3.90$ | $64.41 \pm 0.30$ | $23.20 \pm 4.00$ |
| UNSIR | $86.08 \pm 0.57$ | $91.47 \pm 2.21$ | $50.51 \pm 0.65$ | $59.47 \pm 1.26$ | $49.98 \pm 0.48$ | $64.24 \pm 1.23$ |
| CQMU (ours) | $93.76 \pm 0.57$ | $2.82 \pm 0.54$ | $73.45 \pm 0.94$ | $0.00 \pm 0.00$ | $58.02 \pm 0.45$ | $0.61 \pm 0.13$ |

In Table 4, an ideal MIA Diff is close to zero, and all methods achieve this, with SCRUB slightly higher—likely due to its use of the forgotten data's distribution in the KL objective. MIA Diff does not exhibit a straightforward correlation with $\mathfrak{C}/\mathfrak{M}$ or accuracy, highlighting its role as a complementary metric rather than a replacement for conformal metrics. All methods are significantly faster than retraining

| Metric | MIA Diff. $\downarrow$ | Tsec $\downarrow$ |
|---|---|---|
| RT | $0.12 \pm 0.11$ | $33661.62 \pm 3074.17$ |
| $\nabla\tau$ | $0.00 \pm 0.02$ | $111.19 \pm 0.81$ |
| SCRUB | $1.81 \pm 0.07$ | $104.91 \pm 13.96$ |
| SSD | $0.23 \pm 0.46$ | $568.76 \pm 1.98$ |
| AMN | $0.35 \pm 0.08$ | $234.32 \pm 0.80$ |
| BADT | $0.04 \pm 0.04$ | $9.87 \pm 0.38$ |
| UNSIR | $0.24 \pm 0.07$ | $85.35 \pm 2.09$ |
| CQMU (ours) | $0.02 \pm 0.01$ | $232.07 \pm 0.41$ |

Table 4: Imagenet100 targeted class-wise forgetting with $c, d = 40$, $\alpha = 0.05$, and 10 forgotten class.

(RT); BADT is the quickest (requiring only one iteration), while CQMU is approximately $150\times$ faster than RT, demonstrating its practicality without relying on retrained baselines.

Table 5 compares CQMU with the CPU (fine-tuning) variant and the CR metric (cf. (21)). CQMU consistently meets the desired $\mathfrak{C}/\mathfrak{M}$ targets, achieving a significant $H$ improvement ($\geq 0.13$) over CPU. In contrast, the CR metric can be misleading: smaller prediction sets reduce the denominator, potentially inflating $CR_{\mathcal{V}_f}$ even when true labels are frequently covered (e.g., CQMU's $\mathfrak{M}_{\mathcal{V}_f} = 0.15 > 0.1 = $ CPU's $\mathfrak{M}_{\mathcal{V}_f}$). Similarly, $CR_{\mathcal{V}_r}$ may appear disproportionately small in many-class settings despite high retained

| Metric | CPU | CQMU |
|---|---|---|
| $\mathfrak{C}_{\mathcal{T}_r}(c) \uparrow$ | $1.00 \pm 0.00$ ● | $1.00 \pm 0.00$ ● |
| $\mathfrak{M}_{\mathcal{T}_f}(d) \uparrow$ | $0.02 \pm 0.00$ | $0.06 \pm 0.01$ ● |
| $\mathfrak{C}_{\mathcal{V}_r}(c) \uparrow$ | $0.97 \pm 0.00$ ● | $0.98 \pm 0.00$ ● |
| $\mathfrak{M}_{\mathcal{V}_f}(d) \uparrow$ | $0.10 \pm 0.01$ ● | $0.15 \pm 0.02$ ● |
| $CR_{\mathcal{V}_r}$ | $0.05 \pm 0.00$ | $0.02 \pm 0.00$ |
| $CR_{\mathcal{V}_f}$ | $0.02 \pm 0.00$ | $0.01 \pm 0.00$ |
| $H \uparrow$ | $0.07 \pm 0.01$ | $0.20 \pm 0.02$ |

Table 5: Imagenet targeted class-wise forgetting with $c, d = 100$, $\alpha = 0.05$, and 20 forgotten classes.

coverage (e.g., $\mathfrak{C}_{\mathcal{V}_r} \approx 0.97$). These results verify that *CR does not reliably address fake unlearning*, particularly in scenarios with large label spaces (see Appendix C for a detailed discussion).

We refer the reader to more numerical results in Appendix H.2, including results on CIFAR100, Imagenet100, and 20-NewsGroups and targeted *cluster*-wise forgetting. We discuss the limitations of our framework in Appendix H.3 and conduct a sensitivity analysis in Appendix I.

## 7 CONCLUSION

We have introduced a novel perspective on MU by anchoring it in the framework of CP, enabling rigorous unlearning without the need for retrained baselines. By defining conformal MU and corresponding empirical metrics, we offer a principled approach to evaluate unlearning effectiveness through the exclusion of forgotten data and the retention of coverage for retained data. This framework ensures statistical reliability for unlearning while preserving performance on retained data. The conformal approach is inherently versatile, with potential extensions to regression tasks, graph neural networks, and natural language models. Future work could explore tighter theoretical guarantees, adaptive methods tailored to diverse model architectures, and broader metrics to capture various dimensions of data influence.

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

## A  Notation and Algorithm

### A.1  Notation table

In Table 6 we summarize the important notation used in the paper, especially that introduced in the preliminaries section (§2). Some notation appearing in the main body of the paper might or might not be mentioned in the table, but should be explained in its section when used.

Table 6: Summary of notations.

| Notation | Description | Example / Value |
|---|---|---|
| $\mathcal{X}$ | feature space | e.g., images |
| $\mathcal{Y}$ | label set | e.g., classes |
| $X \in \mathcal{X}$ | feature vector | "dog" image |
| $Y \in \mathcal{Y}$ | label of $X$ | "dog" class |
| $\mathcal{D} \subset \mathcal{X} \times \mathcal{Y}$ | dataset of input–label pairs | whole corpus |
| $p_{X,Y}$ | data distribution | - |
| $\mathcal{D} \sim p_{X,Y}$ | samples drawn identically from $p_{X,Y}$ | $\mathcal{D}_{\text{train}}, \mathcal{D}_{\text{calib}}, \mathcal{D}_{\text{unlearn}}$ |
| $\mathcal{D}_{\text{train}}$ | training set | used to learn $\theta_o$ |
| $\mathcal{D}_{\text{calib}}$ | calibration set for CP | held out from $p_{X,Y}$ |
| $\mathcal{D}_{\text{unlearn}}$ | unlearning set | held out from $p_{X,Y}$ |
| $\mathcal{D}_{\text{test}}$ | testing set | - |
| $W \in \mathcal{W}$ | targeted random variable | class label or projection |
| $\mathcal{W}_{\text{forget}} \subset \mathcal{W}$ | values of $W$ to be forgotten | e.g., certain classes |
| $\mathcal{D}_f$ | forget set | drawn via $p_{X,Y|W}(\cdot \mid W \in \mathcal{W}_{\text{forget}})$ |
| $\mathcal{D}_r$ | retain set | $\mathcal{D}_{\text{unlearn}} \setminus \mathcal{D}_f$ |
| $\theta_o$ | pretrained model parameters | before unlearning |
| $f_{\theta_o}$ | original model | before unlearning |
| $\mathfrak{U}$ | MU algorithm | maps $(\theta_o, \mathcal{D}_{\text{calib}}, \mathcal{D}_{\text{unlearn}}) \mapsto \theta_u$ |
| $\theta_u$ | unlearned model parameters | output of $\mathfrak{U}$ |
| $f_{\theta_u}$ | unlearned model | after forgetting $\mathcal{D}_f$ |
| $\mathfrak{U}_r$ | retraining algorithm | trains from scratch on $\mathcal{D}_r$ |
| $\theta_{\text{RT}}$ | retrained model parameters | output of $\mathfrak{U}_r$ |
| $f_{\theta_{\text{RT}}}$ | retrained model | trained on $\mathcal{T}_r$ only |
| $\mathsf{L}(f_\theta(X), Y)$ | loss / nonconformity measure | cross-entropy, MSE |
| $s(X, Y; \theta)$ | nonconformity score | Eq. (18) |
| $\alpha$ | significance level for conformal | e.g., 0.05 |
| $\hat{q}_\alpha$ | conformal threshold from $\mathcal{D}_{\text{calib}}$ | $\lceil(1-\alpha)(|\mathcal{D}_{\text{calib}}| + 1)\rceil / |\mathcal{D}_{\text{calib}}|$ quantile |
| $C(X)$ | CP set | $\{y : s(X, y; \theta) \leq \hat{q}_\alpha\}$ |
| $Z = (X, Y)$ | generic data pair | e.g., $(X_{\text{test}}, Y_{\text{test}})$ |

### A.2  Our Conformal Machine Unlearning Algorithm

In Algorithm 1, we provide a pseudo-code of the algorithm we use for unlearning. Note that in our experiments, we used the sigmoid surrogate of the indicator function. Since one could use other approximations of the indicator function, we leave the surrogate unspecified in the pseudo-code.

---

**Algorithm 1** Conformal Prediction Machine Unlearning Method (CQMU)

---

1: **Input:** Original model parameters $\theta_o$, forget set $\mathcal{D}_f$, retain set $\mathcal{D}_r$, calibration set $\mathcal{D}_{\text{calib}}$, non-conformity score $s$, error level $\alpha$, margin buffer $\delta$, surrogate steepness $\gamma$, regularization weight $\lambda$, epochs $n$, learning rate $\eta$
2: **Output:** Unlearned model parameters $\theta_u$
3: Initialize $\theta_u \leftarrow \theta_o$
4: **for** $i \leftarrow 1$ **to** $n$ **do**
5:      compute conformal quantile on $\mathcal{D}_{\text{calib}}$

$$\hat{q}_\alpha \leftarrow \min\left\{ t : \frac{1}{|\mathcal{D}_{\text{calib}}|} \sum_{(x,y) \in \mathcal{D}_{\text{calib}}} \mathbf{1}\{s(x,y;\theta_u) \le t\} \ge 1 - \alpha \right\}$$

6:      Identify $(X_q, Y_q)$ whose score is closest to $\hat{q}_\alpha$.
7:      **for** each mini-batch $\mathcal{B}_f \subset \mathcal{D}_f$ and $\mathcal{B}_r \subset \mathcal{D}_r$ **do**
8:          compute surrogate risks over forget/retain sets

$$\hat{\varepsilon}_f(\theta_u) = \frac{1}{|\mathcal{B}_f|} \sum_{(X,Y) \in \mathcal{B}_f} \ell_\gamma((1+\delta)\, p_{\theta_u}(Y \mid X) \; - \; p_{\theta_u}(Y_q \mid X_q)),$$

$$\hat{\varepsilon}_r(\theta_u) = \frac{1}{|\mathcal{B}_r|} \sum_{(X,Y) \in \mathcal{B}_r} \ell_\gamma((1+\delta)\, p_{\theta_u}(Y_q \mid X_q) \; - \; p_{\theta_u}(Y \mid X))$$

9:          define unlearning loss as negative sum of surrogate risks

$$\widetilde{L}(\theta_u) = -[\hat{\varepsilon}_f(\theta_u) + \hat{\varepsilon}_r(\theta_u)]$$

10:      add regularization to keep $\theta_u$ near $\theta_o$

$$J(\theta_u) = \widetilde{L}(\theta_u) + \lambda\, |\theta_u - \theta_o|_2^2$$

11:      gradient update

$$\theta_u \leftarrow \theta_u - \eta\, \nabla_{\theta_u} J(\theta_u)$$

12:      **end for**
13: **end for**
14: **return** $\theta_u$

---

## B  CONFORMAL PREDICTION BACKGROUND

We briefly recall split CP (Angelopoulos and Bates, 2022; Shafer and Vovk, 2008; Foygel Barber et al., 2023). For a trained model $f_\theta$, a (non-)conformity score is

$$s(X, Y; \theta) = \mathsf{L}\big(f_\theta(X), Y\big), \tag{17}$$

where smaller values indicate better conformity. For classification we use

$$s(X, Y; \theta) \triangleq 1 - p_\theta(Y \mid X), \tag{18}$$

with $p_\theta$ the softmax probability (Sadinle et al., 2019). (Other scores apply to regression (Angelopoulos and Bates, 2022), which we do not pursue here.)

Given significance $\alpha$ and calibration set $\mathcal{D}_{\text{calib}}$ of size $n$, let $\hat{q}_\alpha$ be the $\lceil (1-\alpha)(n+1) \rceil / n$ quantile of the calibration scores. For each test point $(X, Y) \in \mathcal{D}_{\text{test}}$ (in our setting, $\mathcal{D}_{\text{test}}$ may be $\mathcal{D}_{\text{unlearn}}$), define the prediction set

$$C(X; \theta) = \big\{ y : \; s(X, y; \theta) \le \hat{q}_\alpha \big\}. \tag{19}$$

If $\mathcal{D}_{\text{calib}}$ and $\mathcal{D}_{\text{test}}$ are exchangeable, split CP guarantees coverage

$$\mathbb{P}(s(X, Y; \theta) \le \hat{q}_\alpha) = \mathbb{P}(Y \in C(X; \theta)) \ge 1 - \alpha, \tag{20}$$

where probability is over the randomness of the calibration set and the test point. In practice, we randomly partition data into $\mathcal{D}_{\text{train}}$, $\mathcal{D}_{\text{calib}}$, and $\mathcal{D}_{\text{unlearn}}$ to preserve exchangeability.

We denote this procedure by $\text{CP}(\theta, s, \alpha)$. Notably, (20) holds for any $\theta$: the guarantee is induced by the calibration procedure rather than the specific model parameters. Our MU objective (see Section 3) exploits this by encouraging small coverage on $\mathcal{D}_f$ (unlearning) and user-specified coverage on $\mathcal{D}_r$ (retention), while respecting the exchangeability assumptions.

## C  RELATED WORK

A dominant view of approximate unlearning is to approach a model retrained from scratch without the forgotten data (Cao and Yang, 2015; Guo et al., 2020; Ginart et al., 2019; Chien et al., 2024; Koloskova et al., 2025). Under this objective, several works provide guarantees or complexity analyses—typically under strong convexity assumptions (Sekhari et al., 2021; Allouah et al., 2025; Thudi et al., 2022). A broad empirical line explores practical mechanisms without formal guarantees, including gradient-influence subtraction (Warnecke et al., 2023; Graves et al., 2021), variational Bayesian updates (Nguyen et al., 2020), teacher–student transfer that expels forgotten labels (Chundawat et al., 2023a), zero-shot noise-based unlearning (Chundawat et al., 2023b), information-theoretic objectives (Foster et al., 2024a; Xu and Strohmer, 2025), KL-divergence–based forgetting (Kurmanji et al., 2023), selective synaptic/gradient dampening for fast updates (Trippa et al., 2024; Foster et al., 2024b), and mixup-driven hard example exposure at the retain–forget boundary (Peng et al., 2025).

To our knowledge, the first conformal-prediction–motivated evaluation for unlearning is due to (Shi et al., 2025), who address *fake unlearning*—misclassified samples whose ground-truth label still appears in the conformal set (cf. Section 1). They define

$$\text{CR}_{\mathcal{D}} \triangleq \frac{\sum_{(x,y)\in\mathcal{D}} \mathbf{1}\{y \in C(x)\}}{\sum_{(x,y)\in\mathcal{D}} |C(x)|}, \tag{21}$$

and target low $\text{CR}$ on $\mathcal{D}_f$ and high $\text{CR}$ on $\mathcal{D}_r$, alongside a wrapper loss

$$L_{\text{unlearn}} = \max(s(X, Y; \theta) - \hat{q}_\alpha, -\Delta),$$

where $s(X, Y; \theta)$ is the conformity score, $\hat{q}_\alpha$ the conformal quantile, and $\Delta$ a buffer.

**Limitations of** $\text{CR}$**.** While offering a conformal lens, $\text{CR}$ can mislead: (i) because the denominator sums set sizes over forgotten samples, $\text{CR}$ may be low even when many forgotten points are still covered; (ii) a few covered retained samples with small sets can inflate $\text{CR}$ even when most retained points are *not* covered (e.g., empty sets); and (iii) in many-class settings (e.g., CIFAR100/ImageNet), large label spaces dominate the denominator, yielding uniformly tiny and hard-to-interpret values (e.g., 0.01 vs. 0.001) that obscure retained–forgotten separation. Empirically, (Shi et al., 2025) sometimes reports $\text{CR}(\mathcal{D}_f)$ higher than or comparable to $\text{CR}(\mathcal{D}_{\text{test}})$ (see Tables 11 and 16 therein), contrary to the intended behavior. Hence, $\text{CR}$ does not robustly resolve fake unlearning or reliably capture unlearning effectiveness.

**Methodological contrast.** The CPU unlearning procedure of (Shi et al., 2025) updates only the conformity scores of forgotten data. In contrast, CQMU operates on both forgotten and retained samples, adds regularization to prevent over-forgetting, and replaces the non-differentiable quantile with a smooth surrogate—the softmax probability of the calibration sample closest to $\hat{q}_\alpha$—yielding more stable optimization. Similar in using external data to induce forgetting, $\nabla\tau$ (Trippa et al., 2024) introduces

$$L_{\text{unlearn}} = \text{ReLU}(L_{\mathcal{V}_f} - L_{\mathcal{D}_f})^2,$$

matching entropy losses between validation and training for forgotten classes to foil an attacker. Unlike CQMU, $\nabla\tau$ neither uses conformity scores nor targets conformal objectives; our results highlight the performance differences between these approaches.

## C.1 How Can Conformal Machine Unlearning Be Generalized?

Given the length constraint of the paper, in this work we focus on MU in classification tasks. However, our work paves the way for a new paradigm of MU based on the rigorous quantification of CP uncertainty. This paradigm can be straightforwardly extended to regression tasks by using proper non-conformity score functions as shown in classical works on CP (cf. Angelopoulos and Bates (2022); Shafer and Vovk (2008)). Moreover, our paradigm could be extended to any field where CP has been explored with proper handling of its notions. For example, multiple works have explored CP in graph neural networks (Zargarbashi et al., 2023; Sadinle et al., 2019; Huang et al., 2023; Clarkson, 2022). Other works have introduced CP for natural language processing (Cha et al., 2025; Quach et al., 2024; Liu et al., 2025; Zhang et al., 2024). This makes our paradigm directly relevant to current research in MU. The extension of our paradigm to these fields is an interesting topic of future research.

# D Proofs of Theoretical Results

## D.1 Proof of Corollary 1

From (20), we have that $\mathbb{P}(Y \in C(X; \theta_u)) \geq 1 - \alpha$. From (4), we obtain

$$\mathbb{P}(Y \in C(X; \theta_u) \,|\, (X, Y) \in \mathcal{D}_f) \leq 1 - \beta \leq 1 - \alpha.$$

Suppose (3) does not hold (i.e. $\mathbb{P}(Y \in C(X; \theta_u) \,|\, (X, Y) \in \mathcal{D}_r) < 1 - \alpha$). Then,

$$\begin{aligned}
\mathbb{P}(Y \in C(X; \theta_u)) &= \mathbb{P}(Y \in C(X; \theta_u) \,|\, (X, Y) \in \mathcal{D}_r)\mathbb{P}((X, Y) \in \mathcal{D}_r) \\
&\quad + \mathbb{P}(Y \in C(X; \theta_u) \,|\, (X, Y) \in \mathcal{D}_f)\mathbb{P}((X, Y) \in \mathcal{D}_f) \\
&< 1 - \alpha,
\end{aligned}$$

a contradiction. Thus, the corollary holds.

## D.2 Proof of Corollary 2

Denote the following events:

$$A \triangleq \{Y \in C(X; \theta_u)\}, B \triangleq \{(X, Y) \in \mathcal{D}_r\}.$$

Assume (20) holds. Then,

$$\begin{aligned}
&P(Y \in C(X; \theta_u)) \geq 1 - \alpha \text{ (by CP)} \\
&P(B)P(A|B) + P(B^c)(P(A|B^c) \geq 1 - \alpha \text{ (by total probability theorem)}.
\end{aligned}$$

Now, assume equation (5) holds. Then,

$$\begin{aligned}
P(B)P(A|B) + P(B^c)(1 - \beta) &\geq 1 - \alpha \\
P(B^c) - \beta P(B^c) + P(B)P(A|B) &\geq 1 - \alpha.
\end{aligned}$$

Since $P(A|B) \leq 1$ and $P(B) = 1 - P(B^c)$ (both by probability axioms),

$$\begin{aligned}
P(B^c) - \beta P(B^c) + 1 - P(B^c) &\geq 1 - \alpha \\
\beta P(B^c) &\leq \alpha.
\end{aligned}$$

But, $\alpha \leq \beta$ (by definition). Hence,

$$\beta P(B^c) \leq \alpha \leq \beta$$

and the proof is complete.

### D.3 Proof of Proposition 1

From Corollary 1, we only need to show that (4) in Definition 1 holds. We have

$$
\begin{aligned}
\beta &\leq \mathbb{P}(s(X_f, Y_f; \theta_u) > \hat{q}_\alpha) \\
&\leq \mathbb{P}(s(X_f, Y_f; \theta_u) \geq s(X_r, Y_r; \theta_u) \mid s(X_r, Y_r; \theta_u) \leq \hat{q}_\alpha) \qquad (22) \\
&\leq \frac{\mathbb{P}(s(X_f, Y_f; \theta_u) \geq s(X_r, Y_r; \theta_u))}{\mathbb{P}(s(X_r, Y_r; \theta_u) \leq \hat{q}_\alpha)} \\
&\leq \frac{\mathbb{P}(s(X_f, Y_f; \theta_u) \geq s(X_r, Y_r; \theta_u))}{1 - \alpha} \qquad (23)
\end{aligned}
$$

where (22) holds due to independence of $\mathcal{D}_f$ from $\mathcal{D}_{\text{calib}}, \mathcal{D}_r$, and (23) follows from (Angelopoulos and Bates, 2022, Theorem D.1). Therefore, we have

$$
\mathbb{P}(s(X_f, Y_f; \theta_u) \geq s(X_r, Y_r; \theta_u)) \geq \beta(1 - \alpha),
$$

and the proof is complete.

### D.4 Proof of Corollary 3

Define the following events:

$$
\begin{aligned}
A &:= \{Y \in C(X; \theta_u)\}, \\
B &:= \{(X, Y) \in \mathcal{D}_r\} \Rightarrow B^{\mathsf{c}} := \{(X, Y) \in \mathcal{D}_f\}, \\
Q &:= \{|C(X; \theta_u)| \leq c\}, \\
M &:= \{|C(X; \theta_u)| \leq d\}.
\end{aligned}
$$

From (3), we have

$$
1 - \alpha \leq \mathbb{P}(A \mid B). \qquad (24)
$$

On the other hand, by the law of total probability, we have

$$
\begin{aligned}
\mathbb{P}(A \mid B) &= \mathbb{P}(A \mid B, Q)\mathbb{P}(Q \mid B) + \mathbb{P}(A \mid B, Q^{\mathsf{c}})\mathbb{P}(Q^{\mathsf{c}} \mid B) \\
&\leq \mathbb{P}(A \mid B, Q) + \mathbb{P}(Q^{\mathsf{c}} \mid B) \\
&\leq \mathbb{P}(A \mid B, Q) + \zeta_c. \qquad (25)
\end{aligned}
$$

Combining (24) and (25), we have

$$
\mathbb{P}(A \mid B, Q) \geq 1 - \alpha - \zeta_c. \qquad (26)
$$

Furthermore, from (4), we have

$$
\beta \leq \mathbb{P}(A^{\mathsf{c}} \mid B^{\mathsf{c}}). \qquad (27)
$$

Again, by the law of total probability, we have

$$
\begin{aligned}
\mathbb{P}(A^{\mathsf{c}} \mid B^{\mathsf{c}}) &= \mathbb{P}(A^{\mathsf{c}} \mid B^{\mathsf{c}}, M)\mathbb{P}(M \mid B^{\mathsf{c}}) + \mathbb{P}(A^{\mathsf{c}} \mid B^{\mathsf{c}}, M^{\mathsf{c}})\mathbb{P}(M^{\mathsf{c}} \mid B^{\mathsf{c}}) \\
&\leq \mathbb{P}(A^{\mathsf{c}} \mid B^{\mathsf{c}}, M) + \mathbb{P}(M^{\mathsf{c}} \mid B^{\mathsf{c}}) \\
&\leq \mathbb{P}(A \mid B^{\mathsf{c}}, M) + \eta_d. \qquad (28)
\end{aligned}
$$

Combining (27) and (28), we have

$$
\mathbb{P}(A^{\mathsf{c}} \mid B^{\mathsf{c}}, M) \geq \beta - \eta_d. \qquad (29)
$$

The proof is now complete.

## E Machine Unlearning Framework Derivation

Corresponding to the case where $c, d = |\mathcal{Y}|$ in (7) and (8), we maximize the following optimization objective:

$$
J(\theta_u) \triangleq \frac{1}{|\mathcal{D}_f|} \sum_{X \in \mathcal{D}_f} \mathbf{1}\{Y \notin C(X; \theta_u)\} + \frac{1}{|\mathcal{D}_r|} \sum_{X \in \mathcal{D}_r} \mathbf{1}\{Y \in C(X; \theta_u)\},
$$

where $s(X, Y; \theta_u) = 1 - \mathbb{P}(f_{\theta_u}(X) = Y)$ with the probability being the softmax probability of the model $f_{\theta_u}$.

**Smooth surrogate for the indicators.** Because the indicator $I\{\hat{q}_\alpha < s\}$ is non-differentiable, we adopt the sigmoid surrogate

$$\ell(u) \triangleq \frac{1}{1 + \exp(-\gamma u)}, \qquad \gamma > 0,$$

and replace the empirical rates by

$$\widetilde{\varepsilon}_r(\theta) \triangleq \frac{1}{|\mathcal{D}_r|} \sum_{X \in \mathcal{D}_r} \ell(\hat{q}_\alpha - s(X, Y; \theta)), \tag{30}$$

$$\widetilde{\varepsilon}_f(\theta) \triangleq \frac{1}{|\mathcal{D}_f|} \sum_{X \in \mathcal{D}_f} \ell(s(X, Y; \theta) - \hat{q}_\alpha). \tag{31}$$

Accordingly,

$$\tilde{J}(\theta) \triangleq \widetilde{\varepsilon}_f(\theta) + \widetilde{\varepsilon}_r(\theta).$$

This objective is what we want to optimize. However, the quantile score $\hat{q}_\alpha$ is found using the quantile function which is non-differentiable. Hence, we find the softmax probability of the calibration point in $\mathcal{D}_{\text{calib}}$ whose conformity score is the closest to $\hat{q}_\alpha$ and use that softmax probability in our loss as follows.

**Explicit soft-max expansion of the penalized objective.** Define the soft-max probability under the unlearned model $f_{\theta_u}$ by

$$p_{\theta_u}(Y \mid X) \triangleq \frac{\exp(f_{\theta_u}(X)_Y)}{\sum_{y' \in \mathcal{Y}} \exp(f_{\theta_u}(X)_{y'})}.$$

Replacing $s(X, Y; \theta) = 1 - p_{\theta_u}(Y \mid X)$ inside the sigmoid surrogate yields

$$\widetilde{\varepsilon}_r(\theta_u) = \frac{1}{|\mathcal{D}_r|} \sum_{(X,Y) \in \mathcal{D}_r} \ell(p_{\theta_u}(Y_q \mid X_q) - p_{\theta_u}(Y \mid X)),$$

$$\widetilde{\varepsilon}_f(\theta_u) = \frac{1}{|\mathcal{D}_f|} \sum_{(X,Y) \in \mathcal{D}_f} \ell(p_{\theta_u}(Y \mid X) - p_{\theta_u}(Y_q \mid X_q)),$$

where $(X_q, Y_q) \in \mathcal{D}_{\text{calib}}$ represents the point whose conformity score is closest to $\hat{q}_\alpha$ and its label.
Substituting these in the penalty form gives the trainable loss

$$\mathsf{L}_{\text{pen}}(\theta_u) = -\widetilde{\varepsilon}_f(\theta_u) - \widetilde{\varepsilon}_r(\theta_u)$$

and the updated parameters are obtained via minimising this loss such that

$$\theta_u = \min_\theta \mathsf{L}_{\text{pen}}(\theta).$$

## F  Conformal Unlearning Beyond Exchangeability

The reference (Foygel Barber et al., 2023) provides a framework to relax the exchangeability requirement and still obtain meaningful coverage bounds. Specifically, when split CP is applied to a calibration set $\mathcal{D}_{\text{calib}}$ that is *not* exchangeable with $\mathcal{D}_{\text{unlearn}}$, let $|\mathcal{D}_{\text{calib}}| = n$ and $\mathcal{D}_{\text{calib}} = \{Z_1, \ldots, Z_n\}$. The coverage bound becomes

$$\mathbb{P}(Y \in C(X; \theta)) \geq 1 - \alpha - \sum_{i=1}^n \widetilde{\omega}_i \cdot d_{TV}\big(s(\boldsymbol{Z}), s(\boldsymbol{Z^i})\big), \tag{32}$$

where $\widetilde{\omega}_i = \frac{\omega_i}{\omega_1 + \cdots + \omega_n + 1}$ for a set of user-defined weights $\{\omega_i\}_{i=1}^n$, $d_{TV}(P, Q)$ denotes the total variation distance between distributions $P$ and $Q$, $s(\cdot)$ is the (non-)conformity score function, $\boldsymbol{Z} = \{Z_1, \ldots, Z_n, Z\}$, and $\boldsymbol{Z^i}$ is the set $\boldsymbol{Z}$ with the $i$th entry swapped with $Z$, i.e., $\boldsymbol{Z^i} = \{Z_1, \ldots, Z_{i-1}, Z, Z_{i+1}, \ldots, Z_n, Z_i\}$. Intuitively, the weights $\omega_i$ can be chosen so that

samples $Z_i$ more similar to the test sample $Z$ receive higher weight. For further details on non-exchangeable conformal prediction, see Foygel Barber et al. (2023).

What matters for us here is the effect of this gap correction on our results, mainly in Corollary 1. Let us define the correction gap for a test sample $Z$ as $g(Z)$. Under non-exchangeability, we have (as stated before)

$$\mathbb{P}(Y \in C(X; \theta)) \geq 1 - \alpha - g(Z). \tag{33}$$

It follows that for Corollary 1 to hold, we have to include the coverage gap into (3) and (4). That is,

**Proposition 2.** *In the targeted forgetting scenario, suppose $\mathbb{P}((X, Y) \in \mathcal{D}_r) > 0$. Moreover, suppose $g(Z) > 0$ (otherwise $\mathcal{D}_{\text{unlearn}}$ and $\mathcal{D}_{\text{calib}}$ will be exchangeable). Then, (4) implies (3) if $\beta' \geq \alpha' \geq \alpha + g(Z)$, where $\alpha'$ and $\beta'$ are to replace $\alpha$ and $\beta$ in (3) and (4), respectively.*

*Proof.* From (33), we have that $\mathbb{P}(Y \in C(X; \theta_u)) \geq 1 - \alpha - g(Z)$. From (4), we obtain

$$\mathbb{P}(Y \in C(X; \theta_u) \,|\, (X, Y) \in \mathcal{D}_f) \leq 1 - \beta' \leq 1 - \alpha'.$$

Suppose (3) does not hold (i.e. $\mathbb{P}(Y \in C(X; \theta_u) \,|\, (X, Y) \in \mathcal{D}_r) < 1 - \alpha'$). Then,

$$\begin{aligned}
&\mathbb{P}(Y \in C(X; \theta_u)) \\
&= \mathbb{P}(Y \in C(X; \theta_u) \,|\, (X, Y) \in \mathcal{D}_r)\mathbb{P}((X, Y) \in \mathcal{D}_r) \\
&\quad + \mathbb{P}(Y \in C(X; \theta_u) \,|\, (X, Y) \in \mathcal{D}_f)\mathbb{P}((X, Y) \in \mathcal{D}_f) \\
&< 1 - \alpha',
\end{aligned}$$

a contradiction if $\alpha' \geq \alpha + g(Z)$. Since also $\beta' \geq \alpha'$ by definition, the proposition holds. $\quad\square$

We can see that even if exchangeability is hard to assume in some applications where there might be some distributional shifts between samples (e.g., mean shift), conformal unlearning is still a valid framework with minor corrections involved in the coverage guarantees.

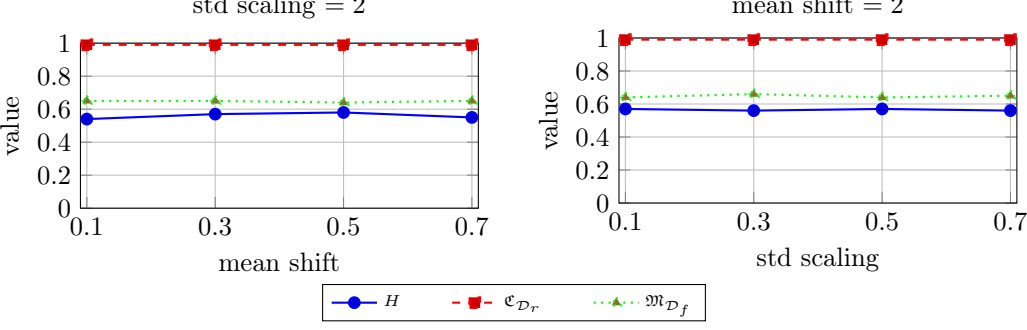

Figure 3: CIFAR100, targeted 10-class forgetting with $\alpha = 0.05$ and $c = d = 100$ (CQMU). The shifts are applied after normalization. Left: metrics vs mean shift with std scaling fixed to 2. Right: metrics vs std scaling with mean shift fixed to 2.

On the technical side, the break in exchangeability can be due to different distributional shifts. For example, Podkopaev and Ramdas (2021) handle shifts in the label distribution, Tibshirani et al. (2019) work on covariate shifts in the given data, and Chernozhukov et al. (2018) focus on dependencies between the data particularly in the time-series case. Generally, as discussed above, the issue is in the proper choice of the weights to account for the distributional shift. Those weights are used to weigh the conformity scores of $\mathcal{D}_{\text{calib}}$ when finding the $(1 - \alpha)$-th quantile. Designing practical conformal unlearning methods to handle different distributional shifts in the data is an extended line of work that is outside the scope of this paper. Meanwhile, we present here the effect of affine transformations on CQMU for mean and standard deviation shifts in the calibration data $\mathcal{D}_{\text{calib}}$ used in unlearning. Fig. 3 shows minute changes in $\mathfrak{C}_{\mathcal{D}_r}$, $\mathfrak{M}_{\mathcal{D}_f}$ and $H$ over the considered shifts. We conclude that CQMU seems **robust to such affine transformations**.

# G  MORE ON THE DATASETS AND THE BASELINES

## G.1  DATASETS

**CIFAR100** (Krizhevsky, 2009) is a carefully curated, labeled subset of the 80 Million Tiny Images dataset developed by Alex Krizhevsky, Vinod Nair, and Geoffrey Hinton . It comprises 60 000 color images of size $32{\times}32$ pixels, evenly distributed across 100 distinct object classes . Each class contains exactly 600 images, which are split into 500 samples for training and 100 for testing. These 100 classes are further organized into 20 higher-level "superclasses," enabling both fine-grained and coarse-grained classification experiments. Every image carries two annotations: a fine label denoting its specific class and a coarse label indicating its superclass. The small $32{\times}32$ resolution makes CIFAR-100 computationally efficient for prototyping convolutional networks and other vision models. Its perfectly balanced class distribution and hierarchical labelling have established CIFAR-100 as a standard benchmark in the computer-vision community. The dataset shares its file-format conventions (Python "pickled" batches, MATLAB files, or binary versions) with CIFAR-10, where each batch bundles image data and labels together (Krizhevsky, 2009). CIFAR-100 is also natively supported in major ML libraries like TensorFlow Datasets and PyTorch's torchvision for seamless integration into research pipelines. In our experiments, we load CIFAR100 using torchvision's datasets library.

**ImageNet100** (Shekhar, 2021) is a compact subset of ILSVRC 2012, containing 100 classes randomly sampled from the original 1,000; it was assembled by the Kaggle user `ambityga` and released in August 2021 as "A Sample of ImageNet Classes." The included categories are listed in a `Labels.json` file, and the dataset is widely used as a smaller, more manageable proxy for ImageNet in research, experimentation, and teaching.

**AG's News Topic Classification Dataset** (Gulli, 2004) selects the four largest categories from a news corpus aggregated by ComeToMyHead (1M+ articles from 2,000+ sources since 2004), yielding 120,000 training samples and 7,600 test samples (30,000/1,900 per class); it was constructed by Xiang Zhang. Each example provides a label (1–4), title, and description; the original release is in CSV, while the Hugging Face version by `sh0416` converts it to JSONL to avoid CSV escaping issues (with automatic Parquet conversion available). The dataset is a standard benchmark for text classification across traditional and modern NLP methods.

**20 Newsgroups Dataset** (Lang, 1995) comprises roughly 20,000 English posts nearly evenly distributed across 20 topics, originally collected by Ken Lang for the 1995 "Newsweeder" study and now a staple benchmark for text classification and clustering. Documents are plain text (headers plus message body), and popular distributions include the scikit-learn version and a SetFit release on Hugging Face that stores `text`, integer `label` (0–19), and `label_text`, providing convenient modern access while preserving the dataset's original structure.

## G.2  ON THE BASELINES

Here we summarize the objectives of each of the baselines and how they achieve unlearning. This section is not meant to be comprehensive but to give a clearer idea about the methods we compare with.

$\nabla\tau$ (Trippa et al., 2024) introduces a new loss objective that focuses on pushing the loss of the data meant to be forgotten from the training data to become larger than the loss of some validation data (carrying the same unlearned labels). They merge it with the original objective of minimizing the loss on the retained data. The new loss is then given by

$$L = \alpha\big(\mathrm{ReLU}(L_{\mathcal{D}_v} - L_{\mathcal{D}_f})\big) + (1 - \alpha)L_{\mathcal{D}_r},$$

where $\alpha$ controls how much emphases should be given to the retained versus forgotten data. In our framework, however, non of the subsets $\mathcal{D}_v$, $\mathcal{D}_f$, and $\mathcal{D}_r$ is seen during training. We assume that this causes the method to fail.

SCRUB (Kurmanji et al., 2023) builds the loss function on the KL-divergence of the unlearning model to a teacher that was trained on the full training data (both $\mathcal{D}_r$ and $\mathcal{D}_f$). This is

nothing but the original base model trained on $\mathcal{D}_{\text{train}}$. They add one more regular loss term to be minimized over the retained data to maintain performance on those points. The final loss that should be minimized becomes

$$L = \frac{\alpha}{N_r} \sum_{x_r \in \mathcal{D}_r} d_{KL}(x_r; \omega^u) + \frac{\gamma}{N_r} \sum_{(x_r,y_r) \in \mathcal{D}_r} l(f(x_r; \omega^u), y_r) - \frac{1}{N_f} \sum_{x_f \in \mathcal{D}_f} d_{KL}(x_f; \omega^u),$$

where $N_r$ is the number of data points to retain, $N_f$ the number of data points to forget, and $\alpha$ and $\gamma$ control the importance of the terms of retaining. Notice that SCRUB tries to make the distributions of the unlearning model and the base model converge to each other on the retained data and diverge from each other on the forgotten data. Hence, when given unseen data, it actually starts learning about the retained data and its accuracy increases on those more than a naively retrained model as shown in the results (e.g., Table 10, Table 9).

SSD (Foster et al., 2024b) uses synaptic dampening of the parameters (weights) of the model which are "specialized" for $\mathcal{D}_f$. SSD compares the "importances" of the weights using the first-order derivative property of the Fisher Information Matrix (FIM) and decides whether to dampen a weight if it is more specialized for $\mathcal{D}_f$ than for other training data, as follows.

$$[]_{\mathcal{D}} = \mathbb{E}\left[ - \frac{\delta^2 \ln p(\mathcal{D} \mid \theta)}{\delta\theta^2}\bigg|_{\theta_D^*} \right],$$

$$[]_{\mathcal{D}} = \mathbb{E}\left[ \left(\frac{\delta \ln p(\mathcal{D} \mid \theta)}{\delta\theta}\right)\left(\frac{\delta \ln p(\mathcal{D} \mid \theta)}{\delta\theta}\right)^T \big|_{\theta_D^*} \right].$$

$$\beta = \min(\lambda \frac{[]_{\mathcal{D},i}}{[]_{\mathcal{D}_{f},i}}, 1),$$

$$\theta_i = \begin{cases} \beta\, \theta_i, & \text{if } []_{\mathcal{D}_{f},i} > \alpha\, []_{\mathcal{D},i}, \\ \theta_i, & \text{if } []_{\mathcal{D}_{f},i} \leq \alpha\, []_{\mathcal{D},i}, \end{cases} \quad \forall\ i \in [0, |\theta|].$$

Generally, they assume that the training data importances can be calculated before training and then the importances of the parameters will be compared between $\mathcal{D}_f$ and $\mathcal{D}_{\text{train}}$. In our framework, $\mathcal{D}_f$ is not part of $\mathcal{D}_{\text{train}}$. Hence, it is not straightforward to argue for a stable relation between the importances. That seems to be the reason why SSD sometimes demolishes the performance of the model (see Table 3).

AMN (Graves et al., 2021) randomly relabels the data to be forgotten. It replaces the classes to be forgotten with new random labels over the whole training set and then retrains the model for a few iterations over the newly labeled data. However, in our case we feed $\mathcal{D}_f$ and $\mathcal{D}_r$ to the unlearning algorithm rather than the full training set. Since $\mathcal{D}_f$ in our framework is not used for training and is just a smaller subset compared to the fraction of forgotten data in the training set, we expect AMN to perform badly on those points.

BADT (Chundawat et al., 2023a) introduces a bad teacher initialized with random noise which induces forgetting by minimizing the KL-divergence between its distribution and that of the unlearned model (student) on the forgotten data. On the other hand, BADT minimizes the divergence between the distribution of the base model and that of the student on the retained data. The objective of BADT is given below.

$$L(x, l_u) = (1 - l_u)\mathcal{KL}(T_s(x)||S(x)) + l_u\mathcal{KL}(T_d(x)||S(x)),$$

where $l_u$ is the label to be forgotten, $x$ is a sample point, $T_s$ is the base model, $T_d$ is the bad teacher model, and $\mathcal{KL}(P, Q)$ is the KL-divergence between the $P$ and $Q$. Note that BADT is suitable for label-wise forgetting since it depends on the label to induce unlearning. That is why we do not use it for cluster-wise forgetting.

UNSIR (Tarun et al., 2024) constructs noisy data by maximizing the loss on the noisy samples that carry the label to be forgotten. Then, it feeds the loss-maximizing noise to the model along with some retrained data in an impair-repair fashion. Similar to BADT, UNSIR depends on the label to be forgotten to construct the loss-maximizing noise. Hence, it is not suitable for targeted cluster-wise forgetting. That is why we do not include its results in those scenarios.

# H EXPERIMENTAL APPENDICES

## H.1 REPRODUCIBILITY DETAILS

**Environment.** Four NVIDIA RTX A5000 GPUs; PyTorch with `nn.DataParallel`; batch size 256; dataloader workers 2; no memory pinning. All vision inputs are normalized with the standard dataset statistics; text tokenization follows the respective model's pipeline.

**Training recipes.** **CIFAR-100:** ResNet18, SGD 50 epochs, initial lr 0.1 with linear decay to $10^{-4}$, momentum 0.9, wd $5 \times 10^{-4}$, cross-entropy, no early stopping. **Tiny-ImageNet (100 labels):** same as CIFAR-100 but 80 epochs. **AG-News, 20 NewsGroups:** BERTa-Distill, 15 epochs, initial lr 0.01, otherwise as above. We report means over 6 seeds.

**Partitions and calibration.** **CIFAR-100:** from the 50k train split, use 45k for training, 5k held out: 2.5k as $\mathcal{D}_{\text{calib}}$ (for quantile estimation during unlearning) and 2.5k split into unseen $\mathcal{V}_f, \mathcal{V}_r$. The 10k test split yields an 8k testing calibration set and label-based $\mathcal{D}_f, \mathcal{D}_r$ for unlearning. Baselines that need validation use $\mathcal{D}_{\text{calib}}$. **Other datasets:** analogous retain/forget and calibration partitions: 90% of the train split is used for training, the remaining 10% of the train split is used for $\mathcal{D}_{\text{calib}}$ (5%) and $\mathcal{V}_r$ and $\mathcal{V}_f$ (5%), and 80% of the test split is used for testing calibration and 20% is used for $\mathcal{D}_r$ and $\mathcal{D}_f$.

**Unlearning optimization.** We retain SGD with the base momentum and weight decay; we tune only the learning rate (grid-search). No scheduler during unlearning.

**Baselines.** $\nabla\tau$ (Trippa et al., 2024), SCRUB (Kurmanji et al., 2023), SSD (Foster et al., 2024b), AMN (Graves et al., 2021), BADT (Chundawat et al., 2023a), UNSIR (Tarun et al., 2024); plus RT on $\mathcal{T}_r$. We use authors' repositories and hyperparameters (for CIFAR-100 from (Foster et al., 2024b; Chundawat et al., 2023a)); we grid-search around released settings to keep compute comparable. Our evaluation applies unlearning on $\mathcal{D}_f/\mathcal{D}_r$.

**MIA evaluation.** For each sample we extract: loss, entropy, prediction margin, logit $L_2$-norm, and top-$k$ probabilities (dynamic $k$). We train a RandomForest attacker with stratified 10-fold cross-validation and report *Adversarial Advantage* (attacker accuracy minus majority-class ratio). We include MIA Diff in all tables.

**Metrics (formal).** $A_{\mathcal{D}}$, $\mathfrak{C}_{\mathcal{D}}(c)$ for $\mathcal{D} \in \{\mathcal{D}_r, \mathcal{T}_r, \mathcal{V}_r\}$, $\mathfrak{M}_{\mathcal{D}}(d)$ for $\mathcal{D} \in \{\mathcal{D}_f, \mathcal{T}_f, \mathcal{V}_f\}$ with $c{=}d$, harmonic mean $H$ over the six conformal metrics (defined as $H = \frac{n}{\sum_i x_i^{-1}}$, with $H{=}0$ if any $x_i{=}0$), MIA Diff, and Tsec. Implementation details for $\mathfrak{C}/\mathfrak{M}$ and calibration protocols follow Sections 2 and 3.

**Code pointers.** Scripts for dataset preparation, partition seeds, hyperparameter grids, and exact command lines are provided in the repository (in the supplementary and will be made public after the paper is published); we fix seeds for data splits and model initialization for full reproducibility. All the reported tables use the same hyperparameters for CQMU: 15 epochs, lr=0.005, $\lambda$=0.03, $\delta$=0.0001, $\gamma = 16$, except that for Table 11 we use lr=0.02.

## H.2 MORE RESULTS

In this appendix we show more results on CIFAR100, Imagenet100, and 20 NewsGroups and both class-wise unlearning as well as cluster-wise unlearning. Note that cluster-wise unlearning is performed by first clustering all the data points in the embedding space using k-means. That is, we take the high representation of the points produced by a pre-trained model (ResNet18) and perform k-means in that space. We use $k = 100$ in k-means resembeling the number of labels in that dataset. Then, we pick a specified number of clusters to forget.

In Fig. 4, we fix the number of forgotten classes and vary $c, d$ (Left) and $\alpha$ (Right). When we vary $c, d$, we expect more points to have larger prediction set size (especially with small $\alpha = 0.05$). This means that the denominators of $\mathfrak{C}$ and $\mathfrak{M}$ increases. Hence, we expect $H$ to

somewhat show some decrease tendancy as $c, d$ increase. We see that behaviour in CQMU, $\nabla\tau$, and AMN. The rest of the methods do not follow a clear trend. However, CQMU attains consistently higher $H$ in general than the rest of the methods. This means that small sets produced by CQMU largely cover the retained data and miscover the forgotten data, as intended. When we vary $\alpha$, we expect $H$ to increase when $\alpha$ increases. The reason is that when $\alpha$ increases, the size of the prediction sets tend to decrease since only a smaller ratio of points are required to be covered by CP. That causes more points to enter the region of interest under $c, d = 40$. As the number of covered retained data points and miscovered forgotten data points increases in this case, $H$ increases. In contrast, when $\alpha$ is small, the size of the prediction set widens to allow it to match the tight coverage bound of CP (e.g. 0.95). We observe that behaviour in CQMU and the rest of the methods except that CQMU maintains high $H$ across different $\alpha$ values. Note that, in practice, we are interested in small $\alpha$ values (typically $0.05 - 0.1$).

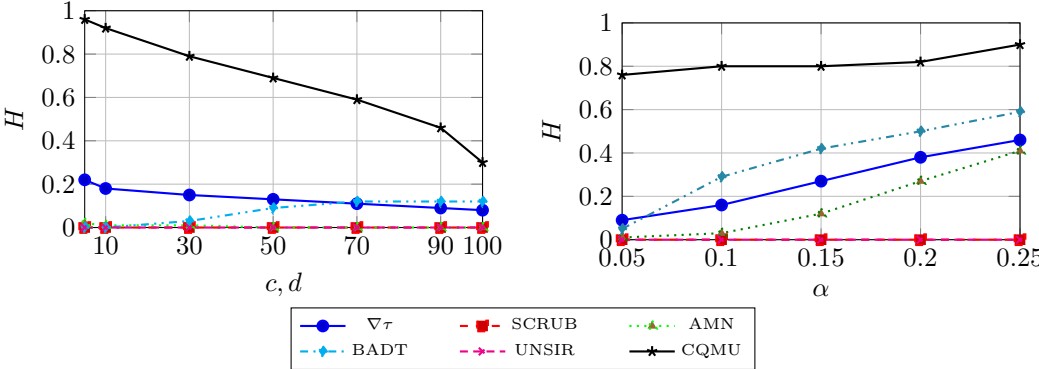

Figure 4: (Left): Plot of $H$ vs. $c = d$ for the different baselines over Imagenet100, $\alpha = 0.05$, and 10 forgotten classes. (Right): Plot of $H$ vs. $\alpha$ for the different baselines over Imagenet100, $c, d = 40$, and 10 forgotten classes.

In all the tables in this appendix, we follow the same convention as in the main text by adding • next to a result if it satisfies the coverage/miscoverage objective. Also, we highlight the best, second, and third best $H$ results with **red** and **blue**, and **cyan**, respectively.

Tables 7 to 11 report more results on the targeted class-wise forgetting. CQMU mostly outperforms the other methods by a good margin in $H$ and attains good $\mathfrak{C}$ and $\mathfrak{M}$ over the given different subsets.

Table 7: CIFAR100 targeted class-wise forgetting with $c, d = 100$, $\alpha = 0.05$, and 20 forgotten class.

| Metric | $\nabla\tau$ | SCRUB | AMN | BADT | UNSIR | CQMU (**Ours**) |
|---|---|---|---|---|---|---|
| $A_{\mathcal{D}_r} \uparrow$ | $76.26 \pm 2.91$ | $99.56 \pm 0.12$ | $100.00 \pm 0.00$ | $45.43 \pm 1.22$ | $44.70 \pm 1.47$ | $72.71 \pm 1.25$ |
| $A_{\mathcal{D}_f} \downarrow$ | $8.64 \pm 2.39$ | $13.39 \pm 1.98$ | $0.00 \pm 0.00$ | $13.39 \pm 3.20$ | $43.99 \pm 0.97$ | $0.00 \pm 0.00$ |
| $A_{\mathcal{T}_r} \uparrow$ | $97.10 \pm 1.60$ | $99.63 \pm 0.12$ | $99.37 \pm 0.10$ | $89.80 \pm 2.43$ | $94.67 \pm 1.18$ | $93.26 \pm 0.22$ |
| $A_{\mathcal{T}_f} \downarrow$ | $31.59 \pm 6.20$ | $77.69 \pm 2.56$ | $65.56 \pm 1.12$ | $49.49 \pm 4.14$ | $94.71 \pm 0.81$ | $1.89 \pm 0.41$ |
| $A_{\mathcal{V}_r} \uparrow$ | $54.97 \pm 1.40$ | $54.69 \pm 0.54$ | $51.39 \pm 0.91$ | $44.05 \pm 1.13$ | $41.95 \pm 1.04$ | $47.40 \pm 0.31$ |
| $A_{\mathcal{V}_f} \downarrow$ | $5.93 \pm 1.97$ | $19.88 \pm 1.42$ | $7.39 \pm 0.56$ | $14.12 \pm 2.42$ | $39.93 \pm 0.95$ | $0.07 \pm 0.15$ |
| $\mathfrak{C}_{\mathcal{D}_r}(c) \uparrow$ | • $1.00 \pm 0.00$ | • $1.00 \pm 0.00$ | • $1.00 \pm 0.00$ | $0.94 \pm 0.00$ | • $0.95 \pm 0.00$ | • $0.99 \pm 0.00$ |
| $\mathfrak{M}_{\mathcal{D}_f}(d) \uparrow$ | • $0.30 \pm 0.03$ | • $0.18 \pm 0.01$ | • $1.00 \pm 0.00$ | • $0.10 \pm 0.03$ | • $0.05 \pm 0.00$ | • $0.34 \pm 0.01$ |
| $\mathfrak{C}_{\mathcal{T}_r}(c) \uparrow$ | • $1.00 \pm 0.00$ | • $1.00 \pm 0.00$ | • $1.00 \pm 0.00$ | • $1.00 \pm 0.00$ | • $1.00 \pm 0.00$ | • $1.00 \pm 0.00$ |
| $\mathfrak{M}_{\mathcal{T}_f}(d) \uparrow$ | $0.02 \pm 0.00$ | $0.02 \pm 0.01$ | $0.01 \pm 0.00$ | $0.01 \pm 0.01$ | $0.00 \pm 0.00$ | • $0.06 \pm 0.00$ |
| $\mathfrak{C}_{\mathcal{V}_r}(c) \uparrow$ | • $0.98 \pm 0.00$ | • $0.96 \pm 0.00$ | • $0.96 \pm 0.00$ | • $0.95 \pm 0.00$ | $0.94 \pm 0.00$ | • $0.99 \pm 0.00$ |
| $\mathfrak{M}_{\mathcal{V}_f}(d) \uparrow$ | • $0.24 \pm 0.02$ | • $0.13 \pm 0.01$ | • $0.13 \pm 0.01$ | • $0.10 \pm 0.02$ | • $0.08 \pm 0.01$ | • $0.23 \pm 0.02$ |
| $H \uparrow$ | $\mathbf{0.11 \pm 0.02}$ | $0.10 \pm 0.03$ | $0.03 \pm 0.00$ | $0.04 \pm 0.04$ | $0.00 \pm 0.00$ | $\mathbf{0.22 \pm 0.01}$ |
| MIA Diff. $\downarrow$ | $2.60 \pm 0.89$ | $3.98 \pm 0.23$ | $3.17 \pm 0.38$ | $2.01 \pm 1.26$ | $5.80 \pm 0.40$ | $0.38 \pm 0.27$ |
| Tsec $\downarrow$ | $16.60 \pm 0.53$ | $36.58 \pm 0.81$ | $10.22 \pm 0.42$ | $1.45 \pm 0.56$ | $14.76 \pm 0.78$ | $23.35 \pm 0.59$ |

Table 8: CIFAR100 targeted class-wise forgetting with $c, d = 20$, $\alpha = 0.05$, and 20 forgotten class.

| Metric | $\nabla\tau$ | SCRUB | AMN | BADT | UNSIR | CQMU (**Ours**) |
|---|---|---|---|---|---|---|
| $A_{\mathcal{D}_r} \uparrow$ | $76.26 \pm 2.91$ | $99.56 \pm 0.12$ | $100.00 \pm 0.00$ | $45.43 \pm 1.22$ | $44.70 \pm 1.47$ | $72.71 \pm 1.25$ |
| $A_{\mathcal{D}_f} \downarrow$ | $8.64 \pm 2.39$ | $13.39 \pm 1.98$ | $0.00 \pm 0.00$ | $13.39 \pm 3.20$ | $43.99 \pm 0.97$ | $0.00 \pm 0.00$ |
| $A_{\mathcal{T}_r} \uparrow$ | $97.10 \pm 1.60$ | $99.63 \pm 0.12$ | $99.37 \pm 0.10$ | $89.80 \pm 2.43$ | $94.67 \pm 1.18$ | $93.26 \pm 0.22$ |
| $A_{\mathcal{T}_f} \downarrow$ | $31.59 \pm 6.20$ | $77.69 \pm 2.56$ | $65.56 \pm 1.12$ | $49.49 \pm 4.14$ | $94.71 \pm 0.81$ | $1.89 \pm 0.41$ |
| $A_{\mathcal{V}_r} \uparrow$ | $54.97 \pm 1.40$ | $54.69 \pm 0.54$ | $51.39 \pm 0.91$ | $44.05 \pm 1.13$ | $41.95 \pm 1.04$ | $47.40 \pm 0.31$ |
| $A_{\mathcal{V}_f} \downarrow$ | $5.93 \pm 1.97$ | $19.88 \pm 1.42$ | $7.39 \pm 0.56$ | $14.12 \pm 2.42$ | $39.93 \pm 0.95$ | $0.07 \pm 0.15$ |
| $\mathfrak{C}_{\mathcal{D}_r}(c) \uparrow$ | $\bullet\, 1.00 \pm 0.00$ | $\bullet\, 1.00 \pm 0.00$ | $\bullet\, 1.00 \pm 0.00$ | $\bullet\, 0.95 \pm 0.01$ | $\bullet\, 0.96 \pm 0.00$ | $\bullet\, 0.99 \pm 0.00$ |
| $\mathfrak{M}_{\mathcal{D}_f}(d) \uparrow$ | $\bullet\, 0.62 \pm 0.07$ | $\bullet\, 0.44 \pm 0.04$ | $\bullet\, 1.00 \pm 0.00$ | $\bullet\, 0.42 \pm 0.11$ | $\bullet\, 0.07 \pm 0.01$ | $\bullet\, 0.96 \pm 0.02$ |
| $\mathfrak{C}_{\mathcal{T}_r}(c) \uparrow$ | $\bullet\, 1.00 \pm 0.00$ | $\bullet\, 1.00 \pm 0.00$ | $\bullet\, 1.00 \pm 0.00$ | $\bullet\, 1.00 \pm 0.00$ | $\bullet\, 1.00 \pm 0.00$ | $\bullet\, 1.00 \pm 0.00$ |
| $\mathfrak{M}_{\mathcal{T}_f}(d) \uparrow$ | $\bullet\, 0.12 \pm 0.07$ | $0.03 \pm 0.01$ | $0.02 \pm 0.00$ | $0.01 \pm 0.01$ | $0.00 \pm 0.00$ | $0.67 \pm 0.04$ |
| $\mathfrak{C}_{\mathcal{V}_r}(c) \uparrow$ | $\bullet\, 0.98 \pm 0.00$ | $\bullet\, 0.96 \pm 0.00$ | $\bullet\, 0.97 \pm 0.00$ | $\bullet\, 0.96 \pm 0.00$ | $0.94 \pm 0.01$ | $\bullet\, 0.98 \pm 0.00$ |
| $\mathfrak{M}_{\mathcal{V}_f}(d) \uparrow$ | $\bullet\, 0.63 \pm 0.07$ | $\bullet\, 0.32 \pm 0.02$ | $\bullet\, 0.43 \pm 0.05$ | $\bullet\, 0.34 \pm 0.11$ | $\bullet\, 0.11 \pm 0.03$ | $\bullet\, 0.88 \pm 0.03$ |
| $H \uparrow$ | $\mathbf{0.41 \pm 0.13}$ | $0.15 \pm 0.04$ | $\mathbf{0.09 \pm 0.02}$ | $0.08 \pm 0.05$ | $0.00 \pm 0.00$ | $\mathbf{0.90 \pm 0.01}$ |
| MIA Diff. $\downarrow$ | $2.60 \pm 0.89$ | $3.98 \pm 0.23$ | $3.17 \pm 0.38$ | $2.01 \pm 1.26$ | $5.80 \pm 0.40$ | $0.38 \pm 0.27$ |
| Tsec $\downarrow$ | $16.60 \pm 0.53$ | $36.58 \pm 0.81$ | $10.22 \pm 0.42$ | $1.45 \pm 0.56$ | $14.76 \pm 0.78$ | $23.35 \pm 0.59$ |

Table 9: ImageNet100 targeted class-wise forgetting with $c, d = 100$, $\alpha = 0.05$, and 10 forgotten class.

| Metric | $\nabla\tau$ | SCRUB | AMN | BADT | UNSIR | CQMU (**Ours**) |
|---|---|---|---|---|---|---|
| $A_{\mathcal{D}_r} \uparrow$ | $77.39 \pm 4.60$ | $89.67 \pm 0.15$ | $100.00 \pm 0.00$ | $64.22 \pm 0.39$ | $50.53 \pm 0.62$ | $73.45 \pm 0.94$ |
| $A_{\mathcal{D}_f} \downarrow$ | $27.84 \pm 4.39$ | $61.93 \pm 1.43$ | $0.00 \pm 0.00$ | $9.28 \pm 3.90$ | $58.90 \pm 1.66$ | $0.00 \pm 0.00$ |
| $A_{\mathcal{T}_r} \uparrow$ | $80.25 \pm 5.25$ | $99.52 \pm 0.00$ | $99.43 \pm 0.02$ | $98.42 \pm 0.43$ | $85.74 \pm 0.84$ | $93.76 \pm 0.57$ |
| $A_{\mathcal{T}_f} \downarrow$ | $48.35 \pm 7.20$ | $99.76 \pm 0.03$ | $71.61 \pm 4.92$ | $51.26 \pm 5.75$ | $91.39 \pm 2.30$ | $2.82 \pm 0.54$ |
| $A_{\mathcal{V}_r} \uparrow$ | $55.55 \pm 3.05$ | $70.19 \pm 0.11$ | $66.91 \pm 0.30$ | $64.41 \pm 0.30$ | $49.67 \pm 0.76$ | $58.02 \pm 0.45$ |
| $A_{\mathcal{V}_f} \downarrow$ | $31.37 \pm 3.87$ | $73.25 \pm 0.42$ | $26.19 \pm 3.49$ | $23.20 \pm 4.00$ | $64.22 \pm 1.34$ | $0.61 \pm 0.13$ |
| $\mathfrak{C}_{\mathcal{D}_r}(c) \uparrow$ | $\bullet\, 0.99 \pm 0.01$ | $\bullet\, 1.00 \pm 0.00$ | $\bullet\, 1.00 \pm 0.00$ | $0.92 \pm 0.01$ | $\bullet\, 0.96 \pm 0.00$ | $\bullet\, 0.98 \pm 0.00$ |
| $\mathfrak{M}_{\mathcal{D}_f}(d) \uparrow$ | $\bullet\, 0.13 \pm 0.04$ | $0.03 \pm 0.01$ | $\bullet\, 0.98 \pm 0.01$ | $\bullet\, 0.26 \pm 0.05$ | $0.03 \pm 0.01$ | $\bullet\, 0.61 \pm 0.04$ |
| $\mathfrak{C}_{\mathcal{T}_r}(c) \uparrow$ | $\bullet\, 0.99 \pm 0.00$ | $\bullet\, 1.00 \pm 0.00$ | $\bullet\, 1.00 \pm 0.00$ | $\bullet\, 1.00 \pm 0.00$ | $\bullet\, 0.98 \pm 0.00$ | $\bullet\, 1.00 \pm 0.00$ |
| $\mathfrak{M}_{\mathcal{T}_f}(d) \uparrow$ | $0.02 \pm 0.01$ | $0.00 \pm 0.00$ | $0.00 \pm 0.00$ | $0.04 \pm 0.03$ | $0.00 \pm 0.00$ | $\bullet\, 0.09 \pm 0.01$ |
| $\mathfrak{C}_{\mathcal{V}_r}(c) \uparrow$ | $\bullet\, 0.96 \pm 0.00$ | $\bullet\, 0.96 \pm 0.00$ | $\bullet\, 0.97 \pm 0.00$ | $\bullet\, 0.97 \pm 0.00$ | $\bullet\, 0.95 \pm 0.00$ | $\bullet\, 0.98 \pm 0.00$ |
| $\mathfrak{M}_{\mathcal{V}_f}(d) \uparrow$ | $\bullet\, 0.09 \pm 0.02$ | $0.03 \pm 0.00$ | $\bullet\, 0.07 \pm 0.01$ | $\bullet\, 0.15 \pm 0.02$ | $0.01 \pm 0.00$ | $\bullet\, 0.24 \pm 0.01$ |
| $H \uparrow$ | $\mathbf{0.08 \pm 0.03}$ | $0.00 \pm 0.00$ | $0.00 \pm 0.00$ | $\mathbf{0.15 \pm 0.07}$ | $0.00 \pm 0.00$ | $\mathbf{0.30 \pm 0.01}$ |
| MIA Diff. $\downarrow$ | $0.00 \pm 0.02$ | $1.81 \pm 0.07$ | $0.35 \pm 0.08$ | $0.04 \pm 0.04$ | $0.25 \pm 0.08$ | $0.02 \pm 0.01$ |
| Tsec $\downarrow$ | $111.19 \pm 0.81$ | $104.91 \pm 13.96$ | $234.32 \pm 0.80$ | $9.87 \pm 0.38$ | $89.65 \pm 4.78$ | $252.07 \pm 0.41$ |

Table 10: ImageNet100 targeted class-wise forgetting with $c, d = 20$, $\alpha = 0.05$, and 10 forgotten class.

| Metric | $\nabla\tau$ | SCRUB | AMN | BADT | UNSIR | CQMU (**Ours**) |
|---|---|---|---|---|---|---|
| $A_{\mathcal{D}_r} \uparrow$ | $77.39 \pm 4.60$ | $89.67 \pm 0.15$ | $100.00 \pm 0.00$ | $64.22 \pm 0.39$ | $50.53 \pm 0.62$ | $73.45 \pm 0.94$ |
| $A_{\mathcal{D}_f} \downarrow$ | $27.84 \pm 4.39$ | $61.93 \pm 1.43$ | $0.00 \pm 0.00$ | $9.28 \pm 3.90$ | $58.90 \pm 1.66$ | $0.00 \pm 0.00$ |
| $A_{\mathcal{T}_r} \uparrow$ | $80.25 \pm 5.25$ | $99.52 \pm 0.00$ | $99.43 \pm 0.02$ | $98.42 \pm 0.43$ | $85.74 \pm 0.84$ | $93.76 \pm 0.57$ |
| $A_{\mathcal{T}_f} \downarrow$ | $48.35 \pm 7.20$ | $99.76 \pm 0.03$ | $71.61 \pm 4.92$ | $51.26 \pm 5.75$ | $91.39 \pm 2.30$ | $2.82 \pm 0.54$ |
| $A_{\mathcal{V}_r} \uparrow$ | $55.55 \pm 3.05$ | $70.19 \pm 0.11$ | $66.91 \pm 0.30$ | $64.41 \pm 0.30$ | $49.67 \pm 0.76$ | $58.02 \pm 0.45$ |
| $A_{\mathcal{V}_f} \downarrow$ | $31.37 \pm 3.87$ | $73.25 \pm 0.42$ | $26.19 \pm 3.49$ | $23.20 \pm 4.00$ | $64.22 \pm 1.34$ | $0.61 \pm 0.13$ |
| $\mathfrak{C}_{\mathcal{D}_r}(c) \uparrow$ | $\bullet\, 1.00 \pm 0.00$ | $\bullet\, 1.00 \pm 0.00$ | $\bullet\, 1.00 \pm 0.00$ | $0.92 \pm 0.01$ | $\bullet\, 1.00 \pm 0.00$ | $\bullet\, 0.97 \pm 0.00$ |
| $\mathfrak{M}_{\mathcal{D}_f}(d) \uparrow$ | $\bullet\, 0.16 \pm 0.07$ | $0.03 \pm 0.01$ | $\bullet\, 0.99 \pm 0.01$ | $0.00 \pm 0.00$ | $0.00 \pm 0.00$ | $\bullet\, 1.00 \pm 0.00$ |
| $\mathfrak{C}_{\mathcal{T}_r}(c) \uparrow$ | $\bullet\, 1.00 \pm 0.00$ | $\bullet\, 1.00 \pm 0.00$ | $\bullet\, 1.00 \pm 0.00$ | $\bullet\, 1.00 \pm 0.00$ | $\bullet\, 1.00 \pm 0.00$ | $\bullet\, 1.00 \pm 0.00$ |
| $\mathfrak{M}_{\mathcal{T}_f}(d) \uparrow$ | $0.03 \pm 0.01$ | $0.00 \pm 0.00$ | $0.00 \pm 0.00$ | $0.01 \pm 0.00$ | $0.00 \pm 0.00$ | $\bullet\, 0.55 \pm 0.02$ |
| $\mathfrak{C}_{\mathcal{V}_r}(c) \uparrow$ | $\bullet\, 0.96 \pm 0.00$ | $\bullet\, 0.95 \pm 0.00$ | $\bullet\, 0.97 \pm 0.00$ | $\bullet\, 0.97 \pm 0.00$ | $\bullet\, 0.99 \pm 0.00$ | $\bullet\, 0.96 \pm 0.00$ |
| $\mathfrak{M}_{\mathcal{V}_f}(d) \uparrow$ | $\bullet\, 0.14 \pm 0.03$ | $0.03 \pm 0.00$ | $\bullet\, 0.17 \pm 0.02$ | $\bullet\, 0.17 \pm 0.03$ | $0.01 \pm 0.00$ | $\bullet\, 0.85 \pm 0.01$ |
| $H \uparrow$ | $\mathbf{0.12 \pm 0.04}$ | $0.00 \pm 0.00$ | $\mathbf{0.01 \pm 0.00}$ | $0.00 \pm 0.00$ | $0.00 \pm 0.00$ | $\mathbf{0.85 \pm 0.01}$ |
| MIA Diff. $\downarrow$ | $0.00 \pm 0.02$ | $1.81 \pm 0.07$ | $0.35 \pm 0.08$ | $0.04 \pm 0.04$ | $0.25 \pm 0.08$ | $0.02 \pm 0.01$ |
| Tsec $\downarrow$ | $111.19 \pm 0.81$ | $104.91 \pm 13.96$ | $234.32 \pm 0.80$ | $9.87 \pm 0.38$ | $89.65 \pm 4.78$ | $252.07 \pm 0.41$ |

Table 11: 20 NewsGroups targeted class-wise forgetting with $c, d = 20$, $\alpha = 0.05$, and 4 forgotten class.

| Metric | $\nabla\tau$ | SCRUB | AMN | BADT | CQMU (**Ours**) |
|---|---|---|---|---|---|
| $A_{\mathcal{D}_r} \uparrow$ | $90.57 \pm 0.74$ | $79.23 \pm 0.55$ | $97.26 \pm 0.06$ | $69.42 \pm 0.18$ | $70.40 \pm 0.00$ |
| $A_{\mathcal{D}_f} \downarrow$ | $5.99 \pm 0.70$ | $24.12 \pm 4.27$ | $0.00 \pm 0.00$ | $32.51 \pm 6.23$ | $0.00 \pm 0.00$ |
| $A_{\mathcal{T}_r} \uparrow$ | $93.83 \pm 0.41$ | $96.42 \pm 0.65$ | $87.12 \pm 0.89$ | $92.16 \pm 0.64$ | $86.03 \pm 0.00$ |
| $A_{\mathcal{T}_f} \downarrow$ | $6.60 \pm 0.91$ | $38.77 \pm 8.54$ | $2.12 \pm 1.02$ | $49.50 \pm 6.75$ | $0.00 \pm 0.00$ |
| $A_{\mathcal{V}_r} \uparrow$ | $74.95 \pm 0.63$ | $74.84 \pm 0.43$ | $70.95 \pm 1.33$ | $71.21 \pm 0.48$ | $68.63 \pm 0.00$ |
| $A_{\mathcal{V}_f} \downarrow$ | $12.31 \pm 1.13$ | $29.28 \pm 5.36$ | $1.40 \pm 1.18$ | $31.46 \pm 4.06$ | $0.00 \pm 0.00$ |
| $\mathfrak{C}_{\mathcal{D}_r}(c) \uparrow$ | • $1.00 \pm 0.00$ | • $0.99 \pm 0.00$ | • $1.00 \pm 0.00$ | $0.91 \pm 0.01$ | • $1.00 \pm 0.00$ |
| $\mathfrak{M}_{\mathcal{D}_f}(d) \uparrow$ | • $0.29 \pm 0.01$ | • $0.21 \pm 0.02$ | • $0.55 \pm 0.04$ | $0.01 \pm 0.00$ | • $0.31 \pm 0.00$ |
| $\mathfrak{C}_{\mathcal{T}_r}(c) \uparrow$ | • $1.00 \pm 0.00$ | • $1.00 \pm 0.00$ | • $1.00 \pm 0.00$ | • $1.00 \pm 0.00$ | • $1.00 \pm 0.00$ |
| $\mathfrak{M}_{\mathcal{T}_f}(d) \uparrow$ | • $0.07 \pm 0.01$ | • $0.10 \pm 0.05$ | • $0.06 \pm 0.01$ | $0.00 \pm 0.00$ | • $0.23 \pm 0.00$ |
| $\mathfrak{C}_{\mathcal{V}_r}(c) \uparrow$ | • $1.00 \pm 0.00$ | • $1.00 \pm 0.00$ | • $0.98 \pm 0.00$ | • $0.97 \pm 0.00$ | • $1.00 \pm 0.00$ |
| $\mathfrak{M}_{\mathcal{V}_f}(d) \uparrow$ | • $0.18 \pm 0.01$ | • $0.12 \pm 0.02$ | • $0.10 \pm 0.01$ | $0.02 \pm 0.00$ | • $0.21 \pm 0.00$ |
| $H \uparrow$ | $\mathbf{0.24 \pm 0.01}$ | $\mathbf{0.24 \pm 0.04}$ | $0.18 \pm 0.02$ | $0.00 \pm 0.00$ | $\mathbf{0.39 \pm 0.00}$ |
| MIA Diff. $\downarrow$ | $2.86 \pm 0.99$ | $3.68 \pm 1.70$ | $0.70 \pm 1.06$ | $2.87 \pm 1.03$ | $3.10 \pm 0.42$ |
| Tsec $\downarrow$ | $19.35 \pm 0.47$ | $35.49 \pm 0.63$ | $25.98 \pm 6.25$ | $8.48 \pm 0.44$ | $45.35 \pm 2.39$ |

Table 12 and Table 13 report results of targeted cluster-wise forgetting. Notice that this scenario is harder for unlearning since the groups still overlap in higher spaces. Nonetheless, CQMU outperforms the other baselines in $\mathfrak{C}$ and $\mathfrak{M}$ in this scenario, and mostly satisfy the coverage and miscoverage bounds as stated in (3) and (4), respectively. More interestingly, we see in both tables that **even RT performs badly in this scenario by highly covering the forgotten data (miscoverage ratio $\leq 10$) in what is called *fake unlearning* (see Section 1)**; this testifies to the rigor of $\mathfrak{C}$ and $\mathfrak{M}$ as metrics to assess MU as well as the fact that **even a retrained model might suffer from fake unlearning when clusters of data are forgotten rather than classes**. Note that BADT and UNSIR are designed to only handle class-wise unlearning, so we did not compare with them.

Table 12: CIFAR100 targeted *cluster*-wise forgetting with $c = d = 100$, $\alpha = 0.1$, and 10 forgotten clusters.

| Metric | RT | $\nabla\tau$ | SCRUB | AMN | CQMU (**Ours**) |
|---|---|---|---|---|---|
| $A_{\mathcal{D}_r} \uparrow$ | $53.01 \pm 0.61$ | $79.07 \pm 3.21$ | $99.78 \pm 0.07$ | $100.00 \pm 0.00$ | $65.42 \pm 3.54$ |
| $A_{\mathcal{D}_f} \downarrow$ | $50.07 \pm 1.63$ | $33.62 \pm 1.50$ | $53.32 \pm 2.40$ | $0.00 \pm 0.00$ | $9.82 \pm 1.57$ |
| $A_{\mathcal{T}_r} \uparrow$ | $99.99 \pm 0.00$ | $98.50 \pm 1.18$ | $99.96 \pm 0.01$ | $99.44 \pm 0.06$ | $91.93 \pm 2.47$ |
| $A_{\mathcal{T}_f} \downarrow$ | $50.10 \pm 0.25$ | $98.88 \pm 1.12$ | $99.98 \pm 0.01$ | $98.03 \pm 0.87$ | $76.73 \pm 4.92$ |
| $A_{\mathcal{V}_r} \uparrow$ | $53.40 \pm 0.39$ | $54.74 \pm 0.80$ | $54.99 \pm 0.56$ | $50.84 \pm 0.34$ | $42.85 \pm 1.64$ |
| $A_{\mathcal{V}_f} \downarrow$ | $50.94 \pm 1.52$ | $52.70 \pm 1.61$ | $54.93 \pm 1.04$ | $45.31 \pm 0.88$ | $30.46 \pm 3.67$ |
| $\mathfrak{C}_{\mathcal{D}_r}(c) \uparrow$ | • $0.90 \pm 0.00$ | • $0.97 \pm 0.01$ | • $1.00 \pm 0.00$ | • $1.00 \pm 0.00$ | • $0.95 \pm 0.00$ |
| $\mathfrak{M}_{\mathcal{D}_f}(d) \uparrow$ | • $0.10 \pm 0.01$ | • $0.36 \pm 0.05$ | • $0.10 \pm 0.01$ | • $1.00 \pm 0.00$ | • $0.77 \pm 0.03$ |
| $\mathfrak{C}_{\mathcal{T}_r}(c) \uparrow$ | • $1.00 \pm 0.00$ | • $1.00 \pm 0.00$ | • $1.00 \pm 0.00$ | • $1.00 \pm 0.00$ | • $0.99 \pm 0.00$ |
| $\mathfrak{M}_{\mathcal{T}_f}(d) \uparrow$ | • $0.10 \pm 0.01$ | $0.00 \pm 0.00$ | $0.00 \pm 0.00$ | $0.00 \pm 0.00$ | • $0.10 \pm 0.02$ |
| $\mathfrak{C}_{\mathcal{V}_r}(c) \uparrow$ | • $0.90 \pm 0.00$ | • $0.90 \pm 0.01$ | • $0.90 \pm 0.00$ | • $0.90 \pm 0.01$ | • $0.91 \pm 0.00$ |
| $\mathfrak{M}_{\mathcal{V}_f}(d) \uparrow$ | • $0.11 \pm 0.01$ | • $0.10 \pm 0.01$ | • $0.12 \pm 0.01$ | • $0.13 \pm 0.02$ | • $0.22 \pm 0.03$ |
| $H \uparrow$ | $\mathbf{0.19 \pm 0.01}$ | $\mathbf{0.01 \pm 0.01}$ | $0.00 \pm 0.00$ | $\mathbf{0.01 \pm 0.00}$ | $\mathbf{0.31 \pm 0.04}$ |
| MIA Diff. $\downarrow$ | $0.28 \pm 0.13$ | $8.07 \pm 1.31$ | $10.17 \pm 0.31$ | $6.27 \pm 1.16$ | $1.29 \pm 1.45$ |
| Tsec $\downarrow$ | $524.62 \pm 3.16$ | $29.39 \pm 0.19$ | $38.78 \pm 4.59$ | $10.92 \pm 0.50$ | $22.91 \pm 0.60$ |

Table 13: CIFAR100 targeted *cluster*-wise forgetting with $c = d = 10$, $\alpha = 0.1$, and 10 forgotten clusters.

| Metric | RT | $\nabla\tau$ | SCRUB | AMN | CQMU (**Ours**) |
|---|---|---|---|---|---|
| $A_{\mathcal{D}_r} \uparrow$ | $53.01 \pm 0.61$ | $79.07 \pm 3.21$ | $99.78 \pm 0.07$ | $100.00 \pm 0.00$ | $64.99 \pm 4.85$ |
| $A_{\mathcal{D}_f} \downarrow$ | $50.07 \pm 1.63$ | $33.62 \pm 1.50$ | $53.32 \pm 2.40$ | $0.00 \pm 0.00$ | $11.58 \pm 0.72$ |
| $A_{\mathcal{T}_r} \uparrow$ | $99.99 \pm 0.00$ | $98.50 \pm 1.18$ | $99.96 \pm 0.01$ | $99.44 \pm 0.06$ | $91.13 \pm 4.07$ |
| $A_{\mathcal{T}_f} \downarrow$ | $50.10 \pm 0.25$ | $98.88 \pm 1.12$ | $99.98 \pm 0.01$ | $98.03 \pm 0.87$ | $76.96 \pm 6.05$ |
| $A_{\mathcal{V}_r} \uparrow$ | $53.40 \pm 0.39$ | $54.74 \pm 0.80$ | $54.99 \pm 0.56$ | $50.84 \pm 0.34$ | $42.16 \pm 2.81$ |
| $A_{\mathcal{V}_f} \downarrow$ | $50.94 \pm 1.52$ | $52.70 \pm 1.61$ | $54.93 \pm 1.04$ | $45.31 \pm 0.88$ | $31.46 \pm 4.10$ |
| $\mathfrak{C}_{\mathcal{D}_r}(c) \uparrow$ | • $0.92 \pm 0.00$ | • $0.99 \pm 0.00$ | • $1.00 \pm 0.00$ | • $1.00 \pm 0.00$ | • $0.95 \pm 0.02$ |
| $\mathfrak{M}_{\mathcal{D}_f}(d) \uparrow$ | $0.08 \pm 0.02$ | • $0.32 \pm 0.04$ | $0.10 \pm 0.01$ | • $1.00 \pm 0.00$ | • $0.79 \pm 0.02$ |
| $\mathfrak{C}_{\mathcal{T}_r}(c) \uparrow$ | • $1.00 \pm 0.00$ | • $1.00 \pm 0.00$ | • $1.00 \pm 0.00$ | • $1.00 \pm 0.00$ | • $0.99 \pm 0.01$ |
| $\mathfrak{M}_{\mathcal{T}_f}(d) \uparrow$ | $0.07 \pm 0.01$ | $0.00 \pm 0.00$ | $0.00 \pm 0.00$ | $0.00 \pm 0.00$ | • $0.13 \pm 0.05$ |
| $\mathfrak{C}_{\mathcal{V}_r}(c) \uparrow$ | • $0.92 \pm 0.01$ | • $0.92 \pm 0.00$ | • $0.93 \pm 0.00$ | • $0.94 \pm 0.00$ | $0.87 \pm 0.04$ |
| $\mathfrak{M}_{\mathcal{V}_f}(d) \uparrow$ | $0.09 \pm 0.02$ | $0.08 \pm 0.01$ | $0.08 \pm 0.01$ | $0.07 \pm 0.02$ | • $0.38 \pm 0.08$ |
| $H \uparrow$ | $\mathbf{0.14 \pm 0.02}$ | $0.00 \pm 0.00$ | $0.00 \pm 0.00$ | $\mathbf{0.01 \pm 0.00}$ | $\mathbf{0.41 \pm 0.09}$ |
| MIA Diff. $\downarrow$ | $0.28 \pm 0.13$ | $8.07 \pm 1.31$ | $10.17 \pm 0.31$ | $6.27 \pm 1.16$ | $1.91 \pm 1.14$ |
| Tsec $\downarrow$ | $524.62 \pm 3.16$ | $29.39 \pm 0.19$ | $38.78 \pm 4.59$ | $10.92 \pm 0.50$ | $23.12 \pm 0.60$ |

### H.3 LIMITATIONS OF CQMU

The limitations of CQMU mainly come from the limitations of CP. We outline them here.

- Split CP requires setting aside a portion of the data exclusively for calibration, which reduces the amount of data available for training and may limit the base model's performance.

- CQMU employs two distinct calibration sets, as described in Section 5. While this enhances its flexibility, it introduces additional complexity in data partitioning and necessitates leaving out more data during training.

- Since CQMU relies on calibration scores to facilitate unlearning, the relationship between the calibration set's score distribution and the unlearning splits' score distributions is critical. This design makes CQMU particularly suited for scenarios involving external data for targeted unlearning. While this is advantageous in cases where the downstream user lacks access to the original training data (e.g., third-party users relying on external datasets with similar characteristics), it can be a limitation when unlearning is performed using internal training data, as CQMU's performance tends to degrade in such cases.

## I SENSITIVITY ANALYSIS

We present analyze the impact of various components in CQMU on the retained-data coverage $\mathfrak{C}_{\mathcal{D}_r}$, the forgotten-data miscoverage $\mathfrak{M}(\mathcal{D}_f)$, and harmonic mean $H$. Experiments were conducted using 1 random seed. We use the same hyperparameters as all the tables (see Appendix H.1).

In Fig. 5, we vary the regularization term $\lambda$. We notice that not a significant change is induced on the forgetting in this case. In fact, $\lambda$ is not directly part of the forgetting loss, it is just a regularization term that is meant to prevent the weights of the unlearning model $\theta_u$ from blowing up. It might be that in other models or settings $\lambda$ affects $H$ more, but this is not what we observe here in $\mathfrak{C}$, $\mathfrak{M}$, or $H$.

In Fig. 6, we vary the CP coverage tolerance level $\alpha$. As in Fig. 4, we expect $H$ to increase as we increase $\alpha$ since it is easier for the model to meet the CP coverage constraint. Except a small drop at $\alpha = 0.2$, the plot confirms that general trend. Notice that $\mathfrak{C}_{\mathcal{D}_r}$ follows a decreasing trend in $\alpha$, that is because the smaller $\alpha$ is, the less coverage level is required on the retained data.

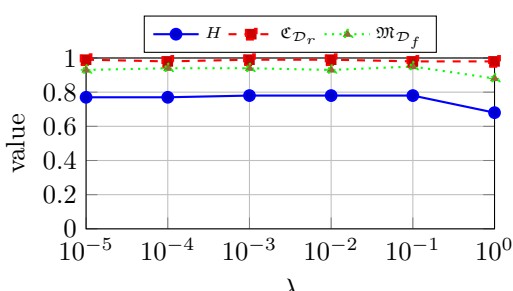

Figure 5: Plots of $\mathfrak{C}_{\mathcal{D}_r}$, $\mathfrak{M}_{\mathcal{D}_f}$, and $H$ vs. the regularization constant $\lambda$.

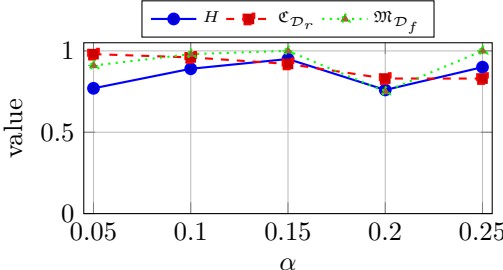

Figure 6: Plots of $\mathfrak{C}_{\mathcal{D}_r}$, $\mathfrak{M}_{\mathcal{D}_f}$, and $H$ vs. the CP coverage tolerance $\alpha$.

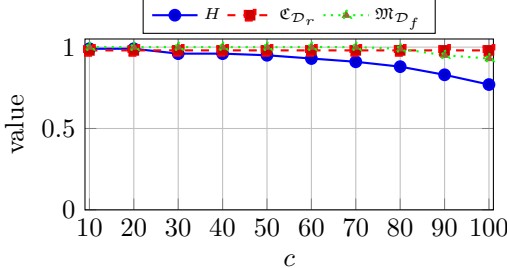

Figure 7: Plots of $\mathfrak{C}_{\mathcal{D}_r}$, $\mathfrak{M}_{\mathcal{D}_f}$, and $H$ vs. the critical set size $c, d$.

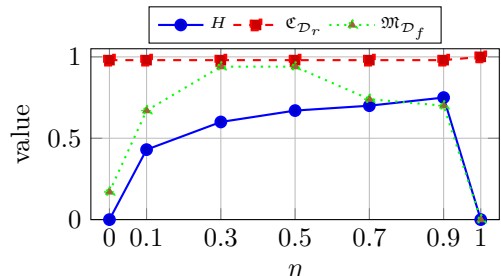

Figure 8: Plots of $\mathfrak{C}_{\mathcal{D}_r}$, $\mathfrak{M}_{\mathcal{D}_f}$, and $H$ vs. forgetting-retaining tradeoff constant $\eta$.

In Fig. 7, we vary the critical set size $c$ (again $c = d$). Similar to *Fig.* 4, we notice that $H$ drops as $c$ increases. As we explained there, this is expected as more points are included in the critical set-size region and hence the denominator of $\mathfrak{C}$ and $\mathfrak{M}$ becomes larger faster than the numerator. However, that behaviour is not clear only by looking at $\mathfrak{C}_{\mathcal{D}_r}$ and $\mathfrak{M}_{\mathcal{D}_f}$ since they remain almost constant. The reason is that the coverage/miscoverage on those subsets does not really change significantly. As the model uses them for unlearning, it overfits to them in terms of unlearning. The drop in $H$ comes from the other terms, especially $\mathfrak{M}_{\mathcal{T}_f}$.

In Fig. 8, we introduce a new forgetting-retaining tradeoff constant $\eta$ that determines how weighs losses in (13). That is, (13) becomes

$$\widetilde{L}(\theta_u; \mathcal{D}_f, \mathcal{D}_r) = -\eta \times \widetilde{\varepsilon}_f(\theta_u) - (1 - \eta) \times \widetilde{\varepsilon}_r(\theta_u). \tag{34}$$

CQMU normally uses equal weight (i.e. $\eta = 0.5$). We notice that when $\eta = 1$, then CQMU becomes similar to CPU since it only gives a quantile-based forgetting term (CPU later adds a loss term from any unleanring method it wraps). In that case, if only forgetting is occurring, the model "overforgets". That is, although one might expect $\mathfrak{M}_{\mathcal{D}_f}$ to become high, that is not the case since the whole model's performance degrades on all data, including the retained data. When $\eta = 0$, the model only retains. Hence, it becomes similar to some form of fine-tuning (i.e. quantile-based finetuning). In that case, we do not expect $\mathfrak{M}_{\mathcal{D}_f}$ to really become high since the performance of the model on the forgotten-data does not drop significantly. This is why balancing the two is important and is a novel formulation of CQMU.

### I.1 Approximate Memory Requirement

CQMU requires approximately 5,574 MB of GPU memory when unlearning 10 classes on CIFAR100, compared to 5,282 MB for $\nabla\tau$, 4,304 MB for SCRUB, 3,054 MB for AMN, 4,024 MB for UNSIR, and 4,032 MB for BADT. CQMU generally consumes relatively a little more memory. This increased memory usage stems from CQMU's approach: it computes quantile scores and simultaneously processes batches from $\mathcal{D}_f$, $\mathcal{D}_r$, and $\mathcal{D}_{\text{calib}}$ during unlearning. Loading all three subsets into GPU memory at once leads to this overhead.

## J  Why Random Instance Forgetting Is Not a Meaningful Unlearning Scenario

In this section, we contend that *random instance forgetting*—the deletion of arbitrarily selected training points—does not reflect meaningful or practical machine unlearning scenarios. Our reasoning is as follows:

1. **Absence of Practical Relevance.** Randomly removing samples does not align with real-world motivations for unlearning. In practice, data is deleted to:

   - Comply with individual privacy requests (e.g., the GDPR "right to be forgotten"), or

- Correct specific errors, such as mislabeled or outdated records (e.g., removing obsolete financial entries).

In both cases, data points are selected based on identity or error, not at random.

2. **Targeted Privacy vs. Randomness.** Fulfilling a data subject's privacy request requires removing *all* data linked to that individual. For instance, if user $u$ requests erasure, we delete $\{(x_i, y_i) \mid \text{owner}(x_i) = u\}$. This process eliminates a structured, potentially correlated subset of the data, rather than a random i.i.d. sample.

3. **Correcting Known Errors.** Similarly, correcting mislabeled samples requires the targeted removal or modification of specific data points. For example, if sample $j$ is identified as mislabeled, we remove or relabel $(x_j, y_j)$ accordingly. Random deletion, on the other hand, may inadvertently eliminate correctly labeled data or fail to address the actual errors.

4. **Statistical Insignificance under i.i.d. Sampling.** Assume the training set $\mathcal{D} = \{(x_i, y_i)\}_{i=1}^n$ is drawn i.i.d. from some distribution $\mathcal{P}$. Deleting $k$ points chosen uniformly at random yields a new empirical distribution

$$\hat{\mathcal{P}}_{-S}(x, y) \;=\; \frac{1}{n-k} \sum_{i \notin S} \delta_{(x_i, y_i)},$$

which converges to the same $\mathcal{P}$ as $n \to \infty$. Thus, random forgetting only perturbs the finite-sample approximation and does not meaningfully alter the learned model's decision boundary beyond sampling noise.

5. **Indistinguishability under Conformal and Accuracy Metrics.** Metrics based on conformal prediction assume exchangeability between the calibration set and the points to be evaluated. However, when the model is trained on the full dataset—including the points to be "forgotten"—this assumption is violated. Even disregarding this, generalization implies that other samples from the same class will occupy similar regions in feature space. As a result, both conformal and accuracy-based metrics are unlikely to detect any meaningful "unlearning effect" when points are removed at random.

6. **Resource Expenditure Without Practical Value.** Assessing unlearning algorithms by evaluating their performance on randomly selected subsets $\mathcal{S} \subset \mathcal{D}$ yields little meaningful information about their effectiveness in real-world scenarios. Specifically, such evaluations do not test the algorithm's ability to:

   - Remove sensitive or personally identifiable information (*targeted forgetting*),
   - Correct specific, known errors (*label unlearning*), or
   - Handle realistic unlearning requests, which typically involve structured or contiguous data subsets.

   As a result, benchmarking on random deletions expends computational resources without providing actionable insights into the algorithm's practical utility.

**Illustrative Example.** Consider a facial-recognition system trained on images from many users. If user Alice requests erasure, we remove all images tagged as `Alice_1.jpg`, `Alice_2.jpg`, etc. Randomly deleting 5 of images (perhaps including some of Alice's) neither guarantees Alice's privacy nor emulates any compliance procedure. Moreover, evaluating unlearning by how often the model still recognizes `Alice_3.jpg` is ill-posed: the model may rely on correlated features learned from other users' faces with similar attributes.

In general, *random instance forgetting* is a strawman: it neither represents the data deletion demands of privacy legislation nor the controlled label correction. Instead, unlearning evaluations should focus on *targeted* forgetting tasks that reflect true operational needs. In a sense, all forgetting scenarios are constrained in the label space $\mathcal{Y}$ or the feature space $\mathcal{X}$ and their extensions. Hence, in their most general forms, the class- and group-wise forgetting scenarios cover all meaningful unlearning requests.

