# OpenReview forum: "On Conformal Machine Unlearning"
_ICLR.cc/2026/Conference — ICLR 2026 Conference Withdrawn Submission_

### Official Review · Reviewer_ossa · 2025-10-17

**Soundness:** 2
**Presentation:** 1
**Contribution:** 2
**Rating:** 2
**Confidence:** 4

**Summary:**

This paper introduces a new definition for machine unlearning based on conformal prediction, providing uncertainty-aware guarantees without the need for the concept of naive retraining. Some comments are provided as follows.

**Strengths:**

1. The paper proposed a new definition for machine unlearning.

2. This paper provides theory and metrics for analyzing and evaluating the proposed method.

**Weaknesses:**

1. First, the most important concern is the reasonability of the defined machine unlearning.

The aim of DP-based unlearning is to ensure the unlearned model is indistinguishable from the retrained model, which is reasonable in both unlearning requirements and privacy protection. However, the definition of (\alpha, \beta) conformal unlearning from the miscoverage level, which is the user-defined miscoverage rate (line 180). It is questionable to assume that users can set the miscoverage level, which has a high influence on model utility. What if the user is malicious? Moreover, how does this definition show that the unlearned model should perform like the model has not "seen" the to-be-forgotten data? The original approximate unlearning shows it by defining the unlearned model indistinguishable from the retrained model.


2. This paper proposes many things to solve three limitations, included in unlearning and unlearning evaluation. However, there lacks a clear problem statement to formulate the main problem the paper solves, which confuses the reviewer about what the key contribution of the paper. Actually, solving and discussing one problem deeply is already enough.

3. The paper wants to reveal a disconnect between accuracy and genuine forgetting and undermining the notion of the retrained model as a universal gold baseline for unlearning. However, in the evaluation, there are still mainly the accuracy-related and MIA metrics. Expected new evaluation metrics or evaluating methods are lacked. And if retraining without the unlearned samples is not unlearning, what is unlearning?

4. Some notations have not been well explained in the paper. For example, what is the mean of "s" in line 172.

5. Equation (20) is used many times in the main text, why not put it in the main text. The main text should be heavily revised as too much important content is put in the appendix.

**Questions:**

See weaknesses.

**Details Of Ethics Concerns:**

No ethics concerns.

---

### Official Review · Reviewer_HdEe · 2025-10-25

**Soundness:** 4
**Presentation:** 2
**Contribution:** 3
**Rating:** 6
**Confidence:** 3

**Summary:**

This paper presents a novel perspective on defining the machine unlearning problem by applying conformal methods and offers an auxiliary statistical evaluation matrix with a direct unlearning method based on this. They purpose to measure the coverage rate “Effective covered frequency(ECF) at c” of the conformal predictor with example smaller than a size c bounded by user defined value to measure on the retained dataset and a complementary matrix “Effective miscovered frequency(EmCF) at d”, on the unlearning dataset to evaluate the unlearning algorithm performance under the conformal method definition. The designed evaluation matrix aims to provide insights into unlearning utility and effectiveness with uncertainty quantification. The paper then proposed a new unlearning method based on the newly defined optimization goal of increasing the conformal prediction bias on retained data and unlearning data. Experiments are followed to provide empirical evidence.

**Strengths:**

1. Novel perspective on machine unlearning with good motivation.
2. A comprehensive, end-to-end framework constituted by a new definition of the problem, a corresponding evaluation matrix, and an unlearning method.
3. Provides statistical, prediction-level guarantees on unlearning effectiveness.

**Weaknesses:**

1. I notice some issues with notation misuse. Specifically, in section 2, D was mislabeled  into D_unlearn = Df ∪Dr. There is also some long-distance formula reference associating with the Formula (20), which is defined in appendix but is frequently referenced in early corollaries.
2. I find the paper lacks discussion of the performance difference between the MIA evaluation and the EmCF at d. The results in Table 4 shows that BADT unlearning methods achieve a comparable unlearning effectiveness with significantly less time spent. This doesn’t correspond to the results shown in Table 2, where the EmCF score at d of BADT is much lower compared to the proposed method. I think more detail should be elaborated on how MIA and EmCF differ in evaluating the unlearning effectiveness.
3. Since the paper claims statistical guarantees, I recommend adding a comparison with a certified method in the experiment section.

**Questions:**

Please demonstrate the trade off between unlearning set size and the achievable convergence and forgetting guarantees.  Does this limit scalability in practice? I am aware that DP-guaranteed unlearning methods have scalability issues in practice due to the privacy budget.

---

### Official Review · Reviewer_DXky · 2025-11-01

**Soundness:** 3
**Presentation:** 2
**Contribution:** 2
**Rating:** 4
**Confidence:** 3

**Summary:**

This paper first point out the current machine unlearning algorithms that rely on accuracy of retain and forget set could be "fake" unlearning as it does really forget the data. Therefore, the paper defines the new machine unlearning and design new algorithm, CQMU backed by theory; and the experimental results show that the proposed method could achieve better unlearning.

**Strengths:**

1. The paper define the new formulation for machine unlearning, rather than simply replying on accuracy on retain and forget set.

2. The proposed method is backed by theory and the empirical results align with the guarantees.

**Weaknesses:**

1. Random instance forgetting could be meaningful and realistic. E.g. when the company uses the clients' data to train a loan approval model, and some clients request to delete their data from the model, these clients might not belong to the same class or share similar features. How do the proposed formulation handle the above scenario.

2. It is hard to understand the experimental setting, if D_f is not a subset of D_train, what is the unlearning request, does it mean that the D_f is the proxy data for D_train?

**Questions:**

1. At line 172, does it imply the unlearning method can only use forget set?

2. There is a recent work on machine unlearning, which incorporate with adversarial training for unlearning, erasing more knowledge surrounding by the forget sample, I wonder how this could affect.
- The Unseen Threat: Residual Knowledge in Machine Unlearning under Perturbed Samples

3. How to map the proposed method work to the normal machine unlearning setting? That is, only training forget/retain and unseen forget/retain? As the model provider will use all training forget/retain to train the model first, the model provider only obligate to assure the data is not used in the model rather than to assure the "unlearning" forget/retain is used or not.

---

### Official Review · Reviewer_vNki · 2025-11-01

**Soundness:** 3
**Presentation:** 3
**Contribution:** 2
**Rating:** 2
**Confidence:** 4

**Summary:**

Overall：
The paper explores the integration of Conformal Prediction (CP) into the domain of machine unlearning. It aims to establish theoretical and empirical foundations for ensuring distribution-free, confidence-calibrated forgetting. The authors propose a novel framework that reformulates the unlearning objective using conformal risk control, deriving a conformal forgetting guarantee based on prediction set coverage.

**Strengths:**

Strengths
1.	The paper provides a clear theoretical link between conformal prediction and machine unlearning, arguing that conformal guarantees can yield interpretable and distribution-agnostic notions of forgetting.
2.	The proposed conformal unlearning formulation is model-agnostic and can be potentially integrated with various unlearning strategies, such as gradient ascent or knowledge distillation–based methods.
3.	The paper is well-written, with coherent logical flow, formal definitions, and intuitive figures that clarify the relationship between conformal coverage and unlearning behavior.

**Weaknesses:**

Weaknesses
1.	Limited novelty in conformal adaptation – while the paper reframes the forgetting objective through conformal coverage, the adaptation itself is relatively straightforward and closely related to the ideas in prior conformal prediction literature. The main contribution lies in reinterpretation rather than methodological innovation.
2.	The “conformal forgetting guarantee” is not rigorously derived as a bound but rather as a heuristic adaptation of prediction set coverage. The connection to formal privacy or stability guarantees is weak.

**Questions:**

See the comments above

---

### Note · Authors · 2025-11-25

I have read and agree with the venue's withdrawal policy on behalf of myself and my co-authors.